# Large Language Models Play StarCraft II: Benchmarks and A Chain of Summarization Approach

**Weiyu Ma**[1,2], **Qirui Mi**[1,2], **Yongcheng Zeng**[1,2], **Xue Yan**[1,2], **Yuqiao Wu**[1,2], **Runji Lin**[1,2], **Haifeng Zhang**[1,2,4], **Jun Wang**[3]

[1] Institute of Automation, Chinese Academy of Sciences, China
[2] School of Artificial Intelligence, University of Chinese Academy of Sciences, China
[3] AI Centre, Department of Computer Science, UCL
[4] Nanjing Artificial Intelligence Research of IA, China

## Abstract

With the continued advancement of Large Language Models (LLMs) Agents in reasoning, planning, and decision-making, benchmarks have become crucial in evaluating these skills. However, there is a notable gap in benchmarks for real-time strategic decision-making. StarCraft II (SC2), with its complex and dynamic nature, serves as an ideal setting for such evaluations. To this end, we have developed TextStarCraft II, a specialized environment for assessing LLMs in real-time strategic scenarios within SC2. Addressing the limitations of traditional Chain of Thought (CoT) methods, we introduce the Chain of Summarization (CoS) method, enhancing LLMs' capabilities in rapid and effective decision-making. Our key experiments included: 1. LLM Evaluation: Tested 10 LLMs in TextStarCraft II, most of them defeating LV5 build-in AI, showcasing effective strategy skills. 2. Commercial Model Knowledge: Evaluated four commercial models on SC2 knowledge; GPT-4 ranked highest by Grandmaster-level experts. 3. Human-AI Matches: Experimental results showed that fine-tuned LLMs performed on par with Gold-level players in real-time matches, demonstrating comparable strategic abilities. All code and data from this study have been made pulicly available at `https://github.com/histmeisah/Large-Language-Models-play-StarCraftII`

## 1 Introduction

Real-time strategy decision-making and long-term planning are critical AI challenges, necessitating rapid, tactical decisions and strategic adaptability over time. StarCraft II (SC2), one of the world's most popular and challenging e-sports, exemplifies these demands through its dynamic gameplay. Players must manage resources, construct bases, and command armies while making quick decisions and adapting their long-term strategies to evolving battlefield conditions. The game's layered gameplay spans economic management, military strategy, and tactical execution, making SC2 a valuable model for AI research, particularly in reinforcement learning (RL). SC2's complexity, real-time nature, and the fact that it is considered one of the hardest games in the world pose significant challenges for AI systems, requiring them to master various aspects of the game simultaneously. Further details about StarCraft II can be found in the appendix 2. Pioneering efforts, exemplified

---

[†]Correspondence to : ⟨haifeng.zhang@ia.ac.cn ⟩, ⟨jun.wang@cs.ucl.ac.uk⟩

by DeepMind's AlphaStar [24], have demonstrated significant advancements in this domain, showcasing AI's growing proficiency in strategic gameplay. With the evolution of LLMs in areas like reasoning, planning, and decision-making, these models have begun to show potential in tasks traditionally dominated by RL approaches[33]. Benchmarks such as AGENTBENCH [15] have been instrumental in evaluating these capabilities in multi-turn, open-ended contexts. However, despite these developments, a specific benchmark for assessing LLMs' capabilities in real-time strategy decision-making and long-term planning in environments like StarCraft II is conspicuously absent.

Therefore, we have chosen SC2 as the benchmark for evaluating the real-time strategy decision-making and long-term planning capabilities of LLMs. Given the lack of language support in existing SC2 environments, we developed TextStarCraft II. Utilizing the python-sc2 framework, TextStarCraft II converts the complex gameplay dynamics of SC2 into an interactive, text-based format. The python-sc2 interface [1] is utilized to convert game data into text, enabling LLM agents to perform macro-strategic actions through language commands. For micro-strategic actions, we implement a rule-based approach akin to that used by OpenAI Five [2], employing predefined Python scripts. This allows LLM agents to engage in competition against the game's built-in AI, other LLM agents, and human players through the execution of these scripted actions. Addressing the challenges posed by the intricate decision-making process in SC2, we propose the Chain of Summarization(CoS) method. This approach enhances the capacity of LLM Agents in processing complex information and making strategic decisions by incorporating single-frame and multi-frame summarization modules, each aiding in understanding the immediate game state and processing sequential data for strategy formulation and decision-making.

In this study, we conduct a thorough exploration of LLMs' application and effectiveness within SC2 via the TextStarCraft II environment. Our experimental framework includes assessing the CoS method, evaluating the performance of proprietary and fine-tuned open-source LLMs in TextStarCraft II, testing real-time human-computer interaction, and using StarCraft II-themed question-answering tasks evaluated by human experts. We also analyze the impact of varying prompts on LLMs, their strategic preferences, and the interpretability of their decision-making processes.

Our contributions are manifold:

- **TextStarCraft II Development:** We present TextStarCraft II, a novel environment that not only enables the evaluation of LLMs in strategic gaming contexts but also supports real-time human-computer interactions.
- **Chain of Summarization Method and Agent:** By introducing the CoS method and releasing an open-source agent, we offer a powerful example and a high-level interface for the community, fostering further development and interaction with TextStarCraft II.
- **Diverse Evaluations of LLMs:** Our extensive evaluations encompass testing LLMs against the game's built-in AI, assessing their understanding of StarCraft II through experts reviews, and conducting human-AI matches. These methodologies underscore LLMs' proficiency in strategic decision-making and their potential for human-like gameplay.

## 2   Background: StarCraft II

StarCraft II (SC2) is a real-time strategy (RTS) game developed by Blizzard Entertainment, renowned for its strategic depth and complexity. It features three distinct playable races: Terrans (humans with advanced technology), Protoss (a psionic species with powerful abilities), and Zerg (a biologically advanced, insectoid race), each with unique units, structures, and gameplay mechanics. The game's core elements include resource management, base construction, unit production, and tactical combat. Players must efficiently gather resources (minerals and vespene gas), construct and expand bases, compose balanced armies, and execute strategic plans while adapting to opponents' moves. This multi-faceted gameplay requires players to make rapid decisions under incomplete information, balancing short-term tactics with long-term strategy. SC2's complexity is further enhanced by its asymmetric design, where each race has distinct strengths and weaknesses, leading to a rich variety of strategies and counter-strategies. The game also incorporates elements of economics, technology progression, and territorial control, making it a comprehensive test of strategic thinking and multitasking ability.

---

[1] https://github.com/BurnySc2/python-sc2

Since its release in 2010, SC2 has been a cornerstone of competitive gaming, featuring prominently in major e-sports tournaments worldwide. The game's professional scene is characterized by high-level play, particularly from South Korean competitors, though it maintains a strong international presence. SC2's strategic complexity has made it a subject of extensive study, not only within the gaming community but also in academic fields such as artificial intelligence and cognitive science. Researchers have utilized SC2 as a testbed for developing and evaluating AI algorithms, particularly in areas of hierarchical planning, multi-agent coordination, and decision-making under uncertainty. The game's rich strategic landscape, combined with its clear victory conditions and quantifiable performance metrics, makes it an ideal environment for benchmarking AI systems against human expertise in complex, real-time domains. Moreover, SC2 has been used to study human cognition, particularly in areas of attention allocation, decision-making under time pressure, and skill acquisition. The game's replay system, which records every action taken by players, provides researchers with a wealth of data for analyzing human performance and strategy evolution over time.

## 3    Related Work

**StarCraft II Full Game AI**: StarCraft AI research, initially focused on StarCraft I with developments like BiCNet [19] for multi-agent coordination, has significantly advanced in the StarCraft II era. The release of PySC2 [23] by DeepMind, coupled with Blizzard's game replays, propelled this research field. A key breakthrough was AlphaStar [24], which achieved Grandmaster level and defeated top players, demonstrating the potential of RL in complex environments.

Subsequent research expanded upon these foundations. Mini-AlphaStar [13] simplified input variables without compromising learning effectiveness. TG [12] and HierNet-SC2 [14] explored efficient RL strategies, with the latter bypassing supervised pre-training. AlphaStar Unplugged [16] represented a leap in offline RL using human replays. TStarBotsX [7] and SCC [26] furthered federated learning approaches, achieving notable success against master and grandmaster level players.

Recent advancements include DI-star [2], which is accessible for home computer deployment, and ROA-Star [9], enhancing AlphaStar's training framework with goal-conditioned exploiters and refined opponent modeling techniques. ROA-Star's practical tests against professional players have shown impressive results, marking significant progress in real-time strategy AI.

**LLM Agent and Benchmark:** The introduction of GPT3.5 [17] has significantly propelled the research on LLM agents forward. Projects such as React [29] and AutoGPT [3] laid the groundwork for more sophisticated implementations. Within the MineDojo environment [5][4], initiatives like GITM [34], Voyager [25], and others [6, 28, 10] have underscored LLM agents' adaptability to a variety of tasks and expansive open-world scenarios. Additionally, environments like TextWorld [4] and ALFWorld [20] enrich agent training by integrating text-based strategy and action execution across simulated and visual contexts, facilitating advanced generalization and adaptive learning. Additionally, optimization methods such as Reinforcement Learning with Human Feedback (RLHF)[31] have also improved the performance of large language models.

Further advancements in multi-agent coordination and virtual social dynamics have been achieved through MetaGPT [8], Camel [11], and Generative Agents [18]. Benchmarking platforms such as AGENTBENCH play a critical role in evaluating these developments, with AGENTBENCH examining decision-making in comprehensive, open-ended contexts.

In StarCraft II, despite the development of advanced AI agents, there remains a gap in evaluating LLMs, especially in real-time strategy and long-term planning. This led to the creation of TextStarCraft II, an environment tailored for testing LLMs in these specific aspects, filling a critical need for natural language interaction capabilities in AI research.

## 4    TextStarCraft II

TextStarCraft II provides a text-based interface for LLMs within the SC2 environment, utilizing the python-sc2 framework to translate complex gameplay into text. Key components include the **Observation-to-Text Adapter** and the **Text-to-Action Adapter**.TextStarCraft II stands out from

---

[2] https://github.com/opendilab/DI-star
[3] https://github.com/Significant-Gravitas/AutoGPT
[4] https://github.com/MineDojo/MineDojo

other text-based environments like TextWorld and ALFWorld due to its more complex and dynamic gameplay. It requires agents to manage multiple aspects such as resource allocation, base building, and military strategy in real-time, challenging both their adaptability and decision-making. Additionally, the environment demands advanced language understanding to interpret and execute more open-ended and diverse commands, enhancing the need for sophisticated natural language processing capabilities. We will introduce the main components of TextStarCraft II below.

**Observation**    TextStarCraft II's observation space is designed to equip LLM agents with essential game insights, effectively navigating the fog of war in StarCraft II. The observations encompass six key categories:

- *Resources:* Game resources and supply levels. ("Minerals: 500, Vespene Gas:200 ")
- *Units:* Types and quantities of the players units.(e.g., "15 Zealots, 5 Carriers")
- *Buildings:* Information on the players buildings.(e.g., "2 Gateway, 1 Nexus")
- *In-Process Activities:* Ongoing construction and production data.(e.g., "Researching Warpgate: 70
- *Enemy Status:* Visible enemy units and buildings
- *Research Progress:* Updates on the player's technological advancements.

This structured approach in the observation space enables LLM agents to efficiently process and utilize vital game data for strategic decision-making in TextStarCraft II.

**Action**    There are mainly two types of actions: *Macro Actions* and *Micro Actions*.

- *Macro Actions:* Covering broad strategic decisions such as Training Units, Building Structures, Researching Technologies, and Other Strategic Maneuvers. Examples include:
  - Building construction (e.g., "Build Cybernetics Core")
  - Unit production (e.g., "Train 5 Probes (Protoss worker)")
  - Research (e.g., "Research Blink Tech")
  - Tactical maneuvers (e.g., "Scout with 1 probe", "Attack enemy base")
- *Micro Actions:* Script-managed for precise placements and targeting, not directly controlled by the agent.

**Reward**    The reward function $\mathcal{R}$ is crucial for aligning agent behavior with the game's objectives, assigning values of $\{-1, 0, 1\}$ based on losing, drawing, or winning a match.

**Game Modes**    TextStarCraft II enriches strategic gameplay with several modes. The **Built-in AI mode** features 10 difficulty levels, ranging from VeryEasy(LV1) to CheatInsane(LV10), and incorporates six strategic styles: RandomBuild, Rush, Timing, Power, Macro, and Air. The **Agent AI mode** enables competition against rule-based AIs and other LLM agents. The **Human mode** supports interactions with real-world players, enhancing gameplay realism.

**Victory Conditions**    In our TextStarCraft2 environment, victory conditions align with those of a standard StarCraft II 1v1 game, akin to benchmarks such as AlphaStar, ROA-Star, or DI-Star. Victory requires the LLM agent to destroy all opponent structures, standardizing our benchmark with practices in various StarCraft II AI competitions.

## 5   Chain of Summarization

The Chain of Summarization (CoS) method, integral to the TextStarCraft II framework, draws inspiration from computer hardware's cache mechanisms and RL's frame skipping techniques. Serving as an enhancement to traditional Chains of Thought (CoT) [27] and as a standard plug-in, CoS refines strategic decision-making in StarCraft II through:

- **Information Compression:** It focuses on key data, reducing overload and sharpening strategic clarity.

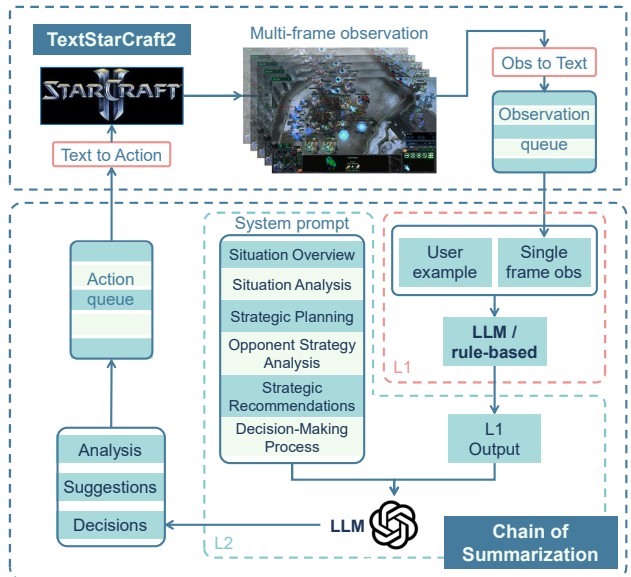

Figure 1: **Interacting with LLM using the Enhanced Chain of Summarization (CoS) Method in TextStarCraft II**. This streamlined LLM-driven gameplay. It begins with initialization, where initial game data is converted to text for processing. Next, Single-Frame and Multi-Frame Summarization refine and summarize observations into actionable insights using advanced LLM reasoning. In Directive Formulation and Action Scheduling, these insights are segmented into specific actions and queued for execution. The process concludes with Action Retrieval and Execution, where actions are implemented in the game. This cycle continually converts new data into text, enhancing LLM performance in the TextStarCraft II.

- **Inference Acceleration:** This approach speeds up decision-making by providing a more comprehensive view of the game's state.
- **Global Understanding:** CoS equips LLMs with a deeper grasp of game strategies, leading to expert-level decisions.

As a versatile tool within the TextStarCraft II framework, CoS can operate both as a standalone plugin, enhancing the environment's utility, and as a direct interface for user interaction with TextStarCraft II. This dual functionality not only showcases CoS as an exemplary model for engaging with our environment but also invites further development and customization by the community, broadening the scope of strategic AI research in gaming. CoS includes Single-Frame Summarization, Multi-Frame Summarization, and Action Extraction for the Action Queue.

**Single-Frame Summarization**    To make TextStarCraft's raw observation data more comprehensible for LLMs, Single-frame Summarization compresses and extracts key information. This process, denoted as $S_{\text{SF}}(\cdot)$, transforms dense TextStarCraft II observations $o$ into a condensed form $\hat{o}$, described by:

$$\hat{o} = S_{\text{SF}}(o). \tag{1}$$

There are two approaches to this compression: a language model-based approach using few-shot learning for better alignment with game rules and a faster, rule-based approach for extraction and filtering. In our experiments, the rule-based approach is primarily used for quicker interactions.

**Multi-Frame Summarization**    Traditional methods query LLMs at each time step for decision-making ([3], [32]). However, this is inefficient for long-duration games like StarCraft II due to high computation costs and slower LLM inference. Our Multi-Frame Summarization method, inspired by caching in computer hardware and frame skipping in RL, addresses these issues. It synchronizes the quick pace of the game with LLM processing, ensuring real-time decision-making efficiency and improved comprehension in complex scenarios. Instead of constant LLM querying, we aggregate

---
**Algorithm 1** Chain of Summarization Interaction in TextStarCraft II
---
**Input**: TextStarCraft II game environment $env$, Chain length $K$

1: Set up the environment and obtain the initial raw observation $o_0 = env.reset()$
2: Initialize the raw observation queue $Q_{\text{obs}}$, action queue $Q_{\text{action}}$ and total reward $R = 0$
3: Add $K$ instances of raw observation $o_0$ to the raw observation queue $Q_{\text{obs}}$
4: **while** $env$ is not terminated **do**
5:    **if** $\text{len}(Q_{\text{obs}}) \geq K$ **then**
6:       Initialize Single-Frame Summarization Queue $Q_{\text{SFS}}$                   ▷ CoS start
7:       **for** raw observation $o$ in $Q_{\text{obs}}$ **do**
8:          Perform Single-Frame Summarization $\hat{o} = S_{\text{SF}}(o)$
9:          Add $\hat{o}$ to the Single-Frame Summarization Queue $Q_{\text{SFS}}$
10:       **end for**
11:       Perform Multi-Frame summarization $\sigma = S_{\text{MF}}(Q_{\text{SFS}})$
12:       Apply Chain of Thought reasoning $\upsilon = CoT(\sigma)$
13:       extract $K$ actions $(a_1, a_2 \cdots, a_K) = Ex(\upsilon)$            ▷ CoS end
14:       Add the $K$ actions $(a_1, a_2 \cdots, a_K)$ to the action queue $Q_{\text{action}}$
15:    **end if**
16:    Obtain the next action $a_t$ from the action queue $Q_{\text{action}}$
17:    Get the reward $r_t$ and the next observation $o_{t+1}$ from the environment $r_t, o_{t+1} = env.step(a_t)$
18:    Add the raw observation $o_{t+1}$ to the raw observation queue $Q_{\text{obs}}$
19:    $R \leftarrow R + r_t$
20: **end while**
21: **return** total reward $R$
---

condensed observation information $\hat{o}$ for $K$ steps into a period summary $\sigma$, described by:

$$\sigma = S_{\text{MF}}(\hat{o}_1, \hat{o}_2, \cdots, \hat{o}_K). \tag{2}$$

This method enables comprehensive analysis and strategic planning through a series of steps, including situation overview, analysis, strategic planning, opponent strategy analysis, suggestion formulation, and decision-making. This process is formalized as $\upsilon$, which is the output of $CoT$ reasoning for summarization $\sigma$, given by:

$$\upsilon = CoT(\sigma). \tag{3}$$

**Action Extraction for Action Queue**   The action queue forms a critical link between the Multi-Frame Summarization results, $\upsilon$, and the TextStarCraft II environment, facilitating communication between the LLM and the game. Within $\upsilon$, key components include analysis, suggestions, and decisions. To convert these into actionable steps, we employ regular expression matching and similarity searching in our action extractor, donated as $Ex(\cdot)$. This process populates the action queue with actions ready for execution in TextStarCraft II. From the output $\upsilon$ of $CoT$ reasoning, we utilize the action extractor to extract $K$ actions, as formalized in:

$$(a_1, a_2, \cdots, a_K) = Ex(\upsilon). \tag{4}$$

The CoS method optimizes decision-making in TextStarCraft II through a streamlined four-stage process: Initially, it sets the initial parameters and transforms the first game frame into text for subsequent analysis. Next, it distills key game observations to provide a concise snapshot of the current situation. Following this, the method translates these summaries into strategic action plans. Finally, it implements the planned actions within the game, thereby completing the decision cycle. This process is depicted in Figure 1. This approach, which updates actions every few frames, effectively manages the fast-paced dynamics of StarCraft II, thereby proving essential for real-time strategic gameplay. The pseudocode is as shown in Algorithm 1.

## 6 Experiment

In our experiment, we detail the setup and key metrics (evaluation metrics detailed in Appendix A.2.) to evaluate macro-strategic decision-making in StarCraft II. We assess the Chain of Summarization's impact on LLM gameplay, compare various LLMs' performance, and evaluate their grasp of StarCraft II strategies. Our experiments also concludes with human-AI interaction tests.

Table 1: Performance of LLMs(Vs LV5) in TextStarCraft II: Comparing models using either the full CoS or CoS without CoT. Evaluation metrics are elaborated in Appendix A.2.

| MODEL | METHOD | WIN RATE | PBR | RUR | APU | TR |
|---|---|---|---|---|---|---|
| USING FULL CoS | | | | | | |
| GPT3.5-TURBO-16K | FULL CoS | 11/20 | 0.0781 | 7875 | 0.7608 | 0.4476 |
| GPT4-TUBOR | FULL CoS | 12/20 | 0.0337 | 8306 | 0.7194 | 0.3452 |
| GEMINI-PRO | FULL CoS | 5/20 | 0.0318 | 9284 | 0.6611 | 0.3571 |
| GLM4 | FULL CoS | 4/20 | 0.0327 | 3131 | 0.6644 | 0.2904 |
| CLAUDE2.1 | FULL CoS | 3/20 | 0.0219 | 10867 | 0.6599 | 0.4312 |
| USING CoS WITHOUT CoT | | | | | | |
| FINETUNE-CHATGLM3 6B | CoS W/O CoT | 3/20 | 0.0528 | 30356 | 0.6547 | 0.1714 |
| FINETUNE-QWEN 1.8B | CoS W/O CoT | 8/20 | 0.0384 | 12826 | 0.7506 | 0.2095 |
| FINETUNE-QWEN 7B | CoS W/O CoT | 9/20 | 0.0421 | 12276 | 0.7234 | 0.3214 |
| FINETUNE-LLAMA2 7B | CoS W/O CoT | 1/20 | 0.0469 | 12295 | 0.5752 | 0.0853 |

## 6.1 Performance Evaluation of Various LLMs

In this section, we assess the performance of closed-source LLMs [21], Llama2 70B [22], and fine-tuned open-source LLMs such as ChatGLM3 6B [30] and Qwen 1.8B [1] in the TextStarCraft II environment against the built-in AI at level 5. The experimental results are shown in Table 1, with the evaluation metrics detailed in Appendix A.2. We tested closed-source LLMs and the un-fine-tuned Llama2 70B using the standard CoS method. The closed-source models performed well, while Llama2 70B could not understand the task requirements and was unable to generate commands based on the given prompts.

Additionally, we fine-tuned open-source models using the entire dataset of GPT3.5-turbo-16k interaction logs with TextStarCraft II. Due to computational resource limitations, we removed the CoT component, keeping only the inputs and outputs. The results showed that all evaluated closed-source LLMs were capable of defeating the level 5 built-in AI. The fine-tuned open-source models, despite a loss in strategic diversity, still managed to overcome the level 5 AI, predominantly adopting a strategy focused on mass-producing stalkers.

Table 2: Performance of models fine-tuned on various datasets in TextStarCraft II, showing win rates(Vs LV5). Datasets are differentiated based on whether they contain all games or only wins, and, for subsets of wins, based on the APU performance percentile.

| DATASET | WIN RATE(VS LV5) |
|---|---|
| FULL DATASET (ALL GAMES) | 28/100 |
| WINS DATASET (ALL WINS) | 48/100 |
| WINS DATASET (BOTTOM 25% APU, 75-100%) | 29/100 |
| WINS DATASET (50-75% APU) | 37/100 |
| WINS DATASET (25-50% APU) | 39/100 |
| WINS DATASET (TOP 25% APU, 0-25%) | 54/100 |

Finally, we investigated the impact of data quality using the Average Population Utilization (APU) metric (detailed in Appendix A.2) to partition the dataset. The results (Table 2) demonstrated that fine-tuning on wins from the top 25% APU games yielded the highest win rate (54/100), while using the full dataset resulted in a lower win rate (28/100). This suggests that training data quality, especially the inclusion of high-performing games, significantly impacts fine-tuned model performance in TextStarCraft II.

## 6.2 Assessing LLMs' Mastery of StarCraft II Concepts

We evaluated the understanding of StarCraft II by LLMs like GPT3.5 and GPT-4, focusing on their knowledge of build orders and game mechanics sourced from prominent StarCraft II forums. De-

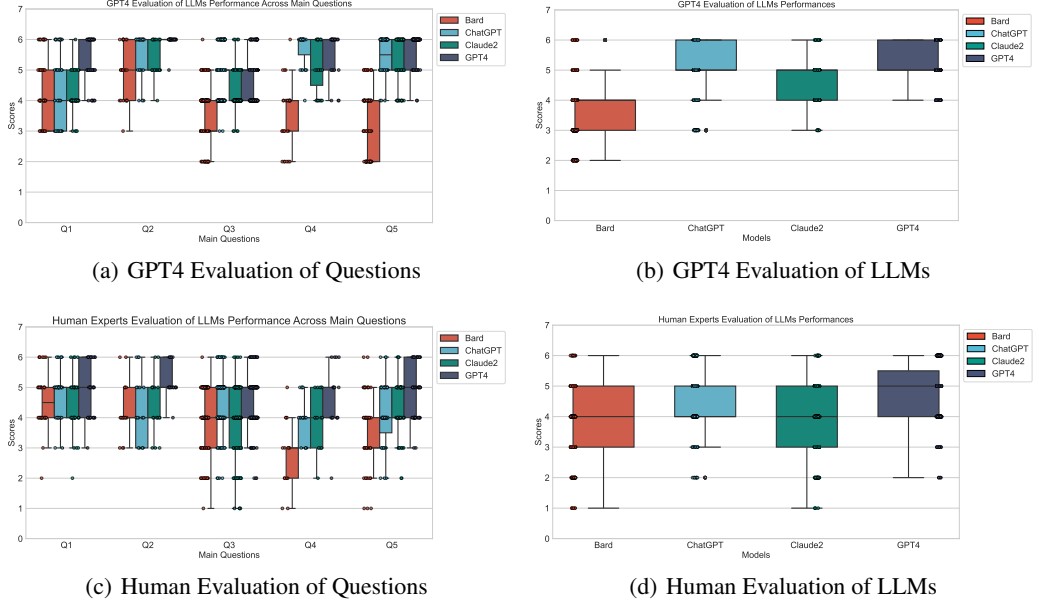

| (a) GPT4 Evaluation of Questions | (b) GPT4 Evaluation of LLMs |
|---|---|
| (c) Human Evaluation of Questions | (d) Human Evaluation of LLMs |

Figure 2: The result of Double-Blind Assessment.

spite a reasonable grasp of basic game dynamics, these models faced challenges with more complex elements like the tech tree and supply constraints.

*Evaluation Methodology:* To gauge the depth of LLMs' StarCraft II knowledge, we tested models, including ChatGPT (GPT3.5), GPT-4, Claude2, and Bard, across five areas: Basic Knowledge, Racial Mechanics, Typical Strategies, Standard Build Orders, and Classic Strategies and Counterplays (detailed in Appendix G). We used a double-blind evaluation, with both Grandmaster-level human experts and GPT-4 assessing the responses to ensure unbiased scoring.

*Evaluation Results:* Both human experts and GPT-4 assessed the LLMs, illustrated in Figures 2 and Table 6, leading to the following insights: In the evaluation of LLMs on a set of complex questions, GPT-4 and ChatGPT demonstrated the best performance, with GPT-4 receiving high scores from both itself and human experts. Claude2 had mixed results, with GPT-4 rating it higher than human experts did. Bard struggled with the complex questions, receiving the lowest scores, especially on questions 3, 4, and 5. Overall, the LLMs were ranked as follows: GPT-4, ChatGPT, Claude2, and Bard. It is worth noting that GPT-4's self-assessment showed more variability compared to the more balanced evaluations provided by human experts.

## 6.3 Human-AI Interaction

Table 3: Match Results: Finetuned Qwen1.8B vs Human.

| HUMAN PLAYER | RANK | MMR | RESULT |
|---|---|---|---|
| PLAYER A | PRO GAMER | 5918 | 0/10 |
| PLAYER B | GRANDMASTER | 5001 | 0/10 |
| PLAYER C | GOLD | 2556 | 5/10 |
| PLAYER D | NEW PLAYER | / | 10/10 |

Following insights from Section 6.1, we evaluated the fine-tuned Qwen1.8B model's real-time interaction with human players on home PCs, requiring only 4GB of GPU memory. This model faced human players of varied skills from the Asian server, including a Grandmaster, a Gold-level player, and a novice, all playing as Zerg against the Protoss-configured LLM agent. Results, detailed in Table 3, show the LLM agent achieving competitive performance, on par with a Gold-level player. This

highlights the LLM's adaptability in strategic play and marks a significant step towards integrating AI into competitive gaming environments on accessible home computing setups.

## 7 Analysis

### 7.1 Impact of Different Prompts

In evaluating the Chain of Summarization method using the GPT3.5-turbo-16k model in TextStarCraft II, we analyzed the effects of two different prompt types on LLM agent performance when playing as Protoss against Zerg. The results, detailed in Table 4, showed a notable improvement in performance with more complex prompts.

Table 4: LLM Agent Win Rates (%) vs. TextStarCraft II AI at Varied Difficulty Levels.

| PROMPT | LV1 | LV2 | LV3 | LV4 | LV5 | LV6 |
|---|---|---|---|---|---|---|
| PROMPT 1 | 87.50 | 66.67 | 25.00 | 12.50 | 0.00 | 0.00 |
| PROMPT 2 | 100.00 | 100.00 | 100.00 | 84.00 | 50.00 | 8.33 |

**Simple Thought Chain:** Using a basic prompt (see Prompt 1), the LLM agent could perform elementary operations like worker production, base establishment, and basic combat unit production. However, this approach was limited in developing advanced strategies, such as research upgrades or comprehensive game analysis, indicating a narrower strategic depth with simpler prompts.

**Complex Thought Chain:** A more intricate prompt (see Prompt 2) guided the LLM agent through a series of critical phases, including situation overview, analysis, strategic planning, and decision-making. This comprehensive approach enabled the agent to engage in advanced strategies like research upgrades, tech tree exploration, and complex military maneuvers. It proved particularly effective against higher difficulty levels (e.g., "Harder" lv 5 ), showcasing enhanced strategic capabilities.

Our analysis underscores the importance of complex prompts that replicate the thought processes of seasoned StarCraft II players. These advanced prompts are essential for LLMs to fully understand and strategically engage in the sophisticated aspects of the game.

### 7.2 Policy Interpretability

Our analysis reveals a stark contrast in decision-making between AlphaStar and our LLM agent. While AlphaStar demonstrates superior micromanagement skills, it occasionally lacks rationality in its strategic choices. Conversely, the LLM agent consistently exhibits logical decision-making, as evidenced by its proactive anticipation of threats and strategic planning, detailed in Appendix E and Appendix F.

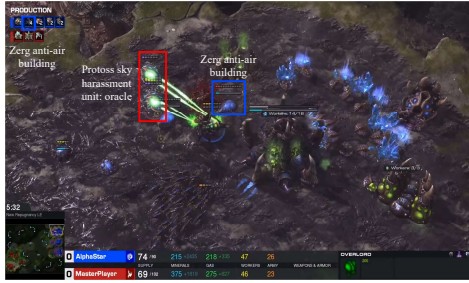

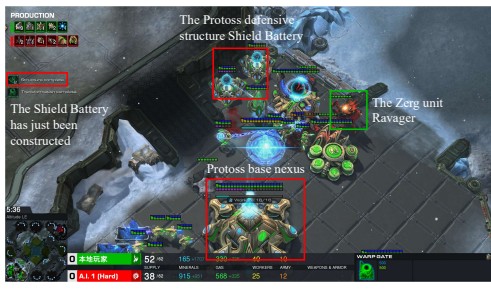

(a) Alphastar failed to construct preemptive defensive structures such as Spore Crawlers to prevent Oracle incursions by the MasterPlayer, resulting in inadequate defensive capabilities against the Oracle harassment.

(b) The LLM agent demonstrated proactive defense measures by constructing Shield Batteries ahead of time, indicating its proactive awareness and preparedness for defensive strategies.

Figure 3: The ability to construct defensive structures and anticipate dangers in advance: Alphastar (left) vs. LLM agent (right).

**Anticipating Threats:** Figure 3 illustrates a stark contrast between AlphaStar and the LLM agent in their ability to anticipate and respond to potential threats. In Figure 3.a, AlphaStar overlooks the impending danger posed by enemy Oracles, failing to recognize the need for adequate defensive structures. This oversight leaves AlphaStar's base vulnerable to a devastating attack, highlighting its inability to proactively assess and mitigate potential risks. Conversely, in Figure 3.b, the LLM agent demonstrates remarkable foresight by analyzing the enemy's strategy and recommending appropriate defensive measures. The agent's log data at 05:00 game time reveals its ability to infer the enemy's Roach-centric strategy based on the presence of key structures such as the Roach Warren and Spawning Pool. Armed with this insight, the LLM agent provides a structured decision-making framework to address the identified threat:

---

- Enemy's Strategy: The enemy has established a Hatchery, Roach Warren, Extractor, and Spawning Pool. This indicates a potential strategy centered around Roach production and early aggression.

- Given the enemy's potential for early aggression with Roaches, we should prioritize defensive structures such as Photon Cannons and Shield Batteries. Additionally, consider scouting the enemy base to gather more information about their strategy.

- Decisions:0: <BUILD PHOTONCANNON> 1: <BUILD SHIELDBATTERY>

---

**Flexible Unit Composition:** AlphaStar often adheres to fixed unit compositions, leading to ineffective strategies in dynamic scenarios (Figure 13.a-d). In contrast, the LLM agent demonstrates adaptability in unit selection, effectively countering varied enemy tactics (Figure 13.e-h). For instance, in Figure 13.f, the Zerg (Built-in AI) employs a Hydra-Roach composition. The LLM agent, playing as Protoss in 09:00, recognizes the need to adapt its strategy based on the enemy's ground-based army. The agent's thought process, as shown in the log, reveals its ability to analyze the situation and make informed decisions:

---

- Enemy's Strategy: The enemy seems to be focusing on a ground-based army composition, consisting of Roaches, Swarm hosts and Hydralisks.

- Key Information: The most important aspect at this moment is our need to expand our unit composition and technology tree to counter the enemy's strategy effectively. We should prioritize unlocking advanced units and upgrades to gain an advantage. Consider researching psionic storm at the Templar Archives to deal with the enemy's ground units effectively.

- Decisions: 4: <RESEARCH PSISTORMTECH>

---

The LLM agent's superior performance in threat anticipation and unit composition adaptability stems from its structured, transparent decision-making process. By analyzing the situation, identifying key information, and making informed decisions based on the evolving game state, the LLM agent demonstrates a blend of human knowledge and logical reasoning. This approach enhances the interpretability of the agent's gameplay, facilitating better collaboration and strategic adaptation in complex scenarios, ultimately leading to more successful outcomes compared to AlphaStar's opaque strategies.

## 8 Discussion

TextStarCraft II enhances the use of LLMs for strategic decision-making in StarCraft II, showcasing their capabilities in adaptive strategy and crisis management. However, the frameworks reliance on rule-based scripts for micro policies, limitation to non-visual data, and a subset of the game's races may restrict the diversity and applicability of AI strategies. Additionally, resource limitations have constrained the performance capabilities of our system. Despite these challenges, TextStarCraft II establishes a new benchmark in RTS games, promoting deeper AI-human collaborative research. Future improvements will focus on integrating visual inputs, expanding race support, and optimizing resource usage to enhance strategic complexity and performance against established AI models.

# 9 Acknowledgements

This work was supported by the National Science and Technology Major Project 2022ZD0116404.

This work was supported by the StarCraft II AI[5] and player communities. We extend our gratitude to Burny[6] for providing detailed documentation and offering warm-hearted replies during the development process. We are grateful to several StarCraft II professional and amateur players who participated in our testing. Special thanks go to:

- Zhiyao(aka. HistTATP)Wang[7] (Champion of China StarCraft II College League, Professional player of SSLT team)
- SLT.Rewhite[8] (Champion of China StarCraft II College League, Professional player of SLT team)
- SLT.JoliwaLoves[9] (Champion of China StarCraft II College League)
- SLT.stargazer[10] (Professional player of SLT team)
- Hist.firmament[11] (Champion of China StarCraft II College Tournament)

Their expertise and insights were invaluable to our research.

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

## Impact Statements

This work introduces a novel method for addressing real-time strategy (RTS) tasks and strategic missions using Large Language Models (LLMs), expanding their application beyond traditional domains. Our approach enhances the interpretability of decisions made by LLMs, offering insights into the language policies that govern AI behavior in complex scenarios. These advancements have potential implications for sectors requiring nuanced decision-making and strategic planning, further bridging the gap between AI capabilities and human-like reasoning in dynamic environments.

## Broader Impact

This paper addresses both the positive and negative societal impacts of employing Large Language Models (LLMs) for real-time strategy gaming environments, such as StarCraft II. On the positive side, enhancing interpretability through language-driven strategies fosters greater understanding and transparency in AI decision-making, which can be beneficial across various sectors that require complex strategic thinking. However, the potential negative impacts include the risk that AI-generated strategies or recommendations might be misleading. This could adversely affect human decision-making, particularly in scenarios where users rely heavily on AI suggestions without sufficient oversight or understanding. Such misleading advice could lead to suboptimal or even harmful decisions in critical areas like security or management. To mitigate these issues, we advocate for rigorous validation of AI advice, comprehensive user education on the AI system's capabilities and limitations, and the development of fail-safes that can alert users to potential inaccuracies or biases in AI-generated strategies.

## Code Of Ethics

This research fully adheres to the NeurIPS Code of Ethics. We have ensured that all experiments, data handling, and dissemination of results maintain high ethical standards and transparency. The methodologies and technologies used in the study are applied in a controlled environment with no real-world deployment, mitigating the risk of harm. Additionally, we have preserved the anonymity of any data used during the training and testing phases to comply with privacy and ethical guidelines.

# A Experiment Setup and Metrics Explanation

## A.1 Experiment Setup

**Agent and Opponent Selection:** To ensure a consistent and controlled experimental environment, LLM agents are configured to play as Protoss against Zerg AI opponents. This setup allows for a systematic evaluation of strategic capabilities across a range of difficulties. The difficulty levels are defined as follows to cover a comprehensive spectrum of challenges:

| Level | 1 | 2 | 3 | 4 | 5 |
|---|---|---|---|---|---|
| **BLZ Difficulty** | Very Easy | Easy | Medium | Hard | Harder |
| **Level** | 6 | 7 | 8 | 9 | 10 |
| **BLZ Difficulty** | Very Hard | Elite | Cheat Vision | Cheat Money | Cheat Insane |

Table 5: AI Difficulty Levels for StarCraft II

**Map Selection:** For our experiments, we selected the Altitude LE and Ancient Cistern LE maps from the 2023 StarCraft II esports 1v1 ladder. These maps provide a variety of strategic challenges that are representative of current competitive play. Additional details about these maps can be found on their respective Liquipedia pages: `https://liquipedia.net/starcraft2/Altitude_LE` and `https://liquipedia.net/starcraft2/Ancient_Cistern_LE`.

**Parameter Settings:** The temperature parameter is fixed at 0.1 to focus on strategy-driven actions over randomness.

**Game Version:** Our experiments were conducted across three different versions of the game to ensure robustness and applicability of the results. The versions tested include Patch 5.0.11, Patch 5.0.12, and Patch 5.0.13.

## A.2 Evaluation Metrics:

Building on StarCraft II's established player performance analytics, our evaluation framework in TextStarCraft II integrates these insights with custom adaptations to comprehensively assess LLM agents gameplay strategies.

**Win Rate:** The most crucial metric, Win Rate directly reflects the agent's capability and performance within the environment. It is calculated as the percentage of games won by the agent out of the total games played.

**Population Block Ratio (PBR):** PBR is a measure of the agents macro-management effectiveness, particularly in resource allocation and population growth. It is defined as:

$$\text{PBR} = \frac{\text{Time at Population Cap}}{\text{Game Duration}} \tag{5}$$

Here, the metric calculates the ratio of time spent at population capacity to the total game time until the agent first reaches maximum supply (200/200). A higher PBR indicates poorer macro-strategic decision-making and ineffective planning.

**Resource Utilization Ratio (RUR):** RUR evaluates the agent's efficiency in resource management throughout the game. It is calculated as:

$$\text{RUR} = \frac{\text{Total Minerals + Total Gas Used}}{\text{Game Duration}} \tag{6}$$

This metric assesses the total resources used against the duration of the game until the agent first reaches maximum supply. A higher RUR signifies underutilization of resources, reflecting weaker macro-strategic capabilities.

**Average Population Utilization (APU):** APU measures the efficiency in utilizing the available population capacity. It is calculated as:

$$\text{APU} = \frac{1}{N} \sum_{i=1}^{N} \left( \frac{\text{Used Population at } i^{th} \text{ step}}{\text{Population Cap at } i^{th} \text{ step}} \right) \tag{7}$$

The metric averages the ratio of used population to population cap until the agent first reaches maximum supply. A higher APU suggests better utilization of population and more effective macro-management.

**Technology Rate (TR):** TR assesses the extent to which the agent explores and utilizes the technology tree. It is defined as:

$$TR = \frac{\text{Completed Technologies}}{\text{Total Technologies Available}} \tag{8}$$

This metric calculates the proportion of completed technologies and buildings to the total available, from the start to the end of the game. TR indicates the agents tendency towards technological advancement and is not necessarily indicative of better or worse performance.

**Data Collection and Analysis:** Due to technical constraints with the sc2reader library for AI-generated replays, we utilized gameplay log data for in-depth analysis. This approach ensures a thorough and precise evaluation based on our tailored metrics.

## A.3 Compute Resource

**Training Phase**: To fine-tune open-source Large Language Models, we utilized two NVIDIA A100 40GB GPUs, providing the necessary computational power for extensive model training. The entire experiment took approximately 70 hours.

**Testing Phase**: The development and testing of these fine-tuned models were performed using two NVIDIA A100 40GB GPUs. For running the StarCraft II environment, we employed an NVIDIA GeForce RTX 3060 GPU paired with a 13th Gen Intel(R) Core(TM) i5-13400F processor operating at 2.50 GHz, ensuring smooth execution of the game simulations.

## A.4 Human Experts

In the course of our research, we engaged 30 StarCraft II masters and grandmasters, including professional and semi-professional players, to participate in our experiments. To safeguard their privacy, we have chosen not to disclose their personal information in this paper.

Each participating player was compensated with 15 USD for their involvement in the study.

# B    Experiment Figure

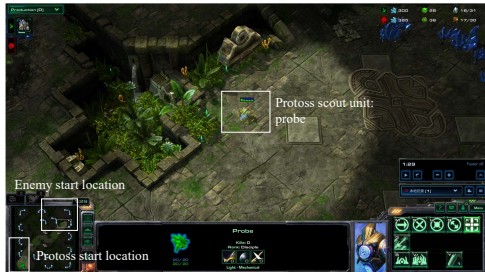

(a) In the early game stage, scouting is crucial. The Protoss have decided to dispatch a Probe to scout the enemy's main base.

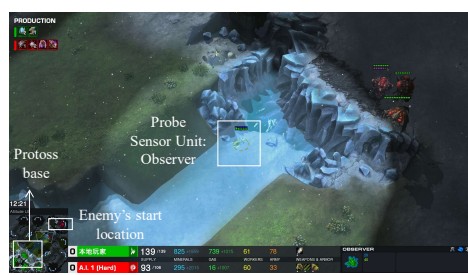

(b) The LLM agent values enemy scouting. The Protoss Observer spotted Brood Lords and advanced tech in the Zerg's third base.

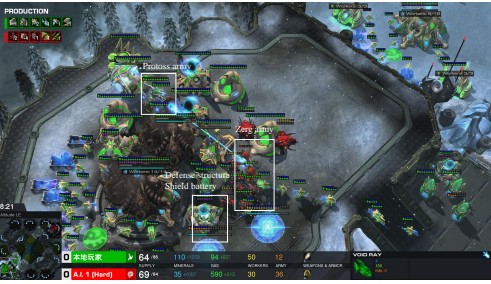

(c) The Protoss employs Shield Batteries as a pivotal defensive strategy against Zerg aggression.

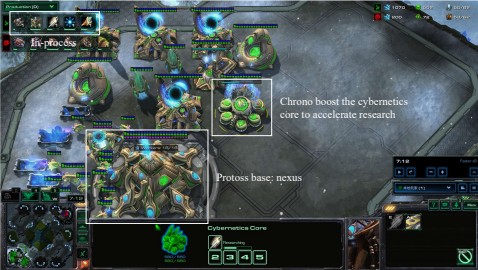

(d) The Protoss are utilizing the Chronoboost ability to accelerate the "Protoss Air Weapon Level 1" upgrade in the Cybernetics Core.

Figure 4: Early and mid-game snapshots of the Protoss (LLM agent), emphasizing defense and scouting.

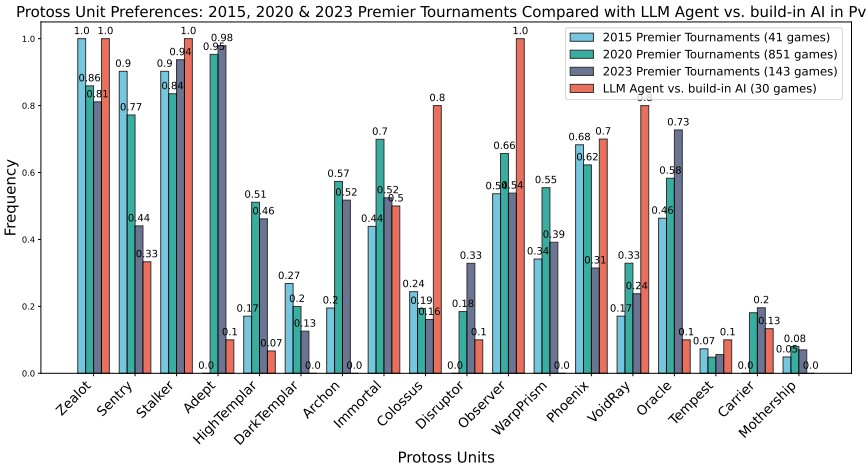

Figure 5: Comparison of Unit Preferences: Analysis of LLM Agent in 30 Games vs. Professional Players in Esports Tournaments from 2015, 2020, and 2023

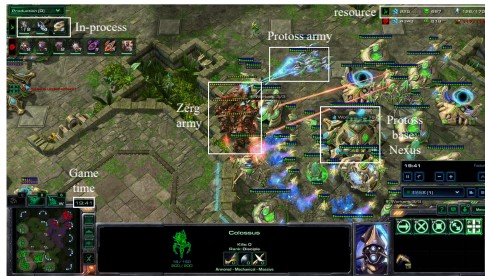

(a) The Protoss strategically transitions to a Void Ray-centric composition, adept at countering the might of Zerg's Ultralisks.

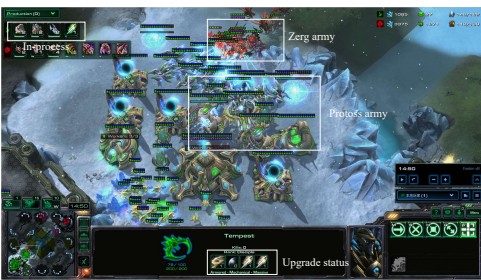

(b) Protoss defended against Zerg's third wave with phoenixes, tempests, and voidrays.

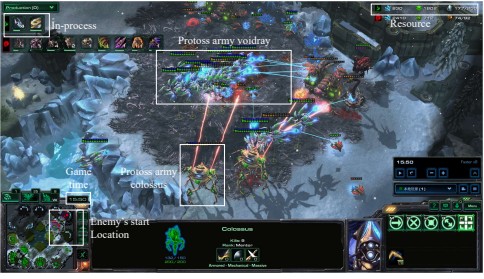

(c) The Protoss, employing the classic combination of Colossus and Voidray units, successfully decimated the Zerg's expansion base.

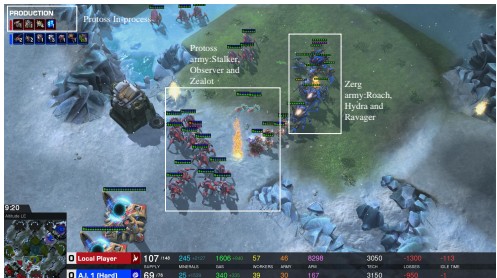

(d) The LLM agent uses stalkers, zealots and observers to counter the Harder AI's attacks, gradually shifting towards a powerful air unit and Psionic Storm strategy.

Figure 6: Mid to late-game visuals of the Protoss (LLM agent), showcasing mixed-unit assembly and troop transitioning.

Table 6: Statistical Evaluation of Models

| Model | Mean Score | | Std Dev | | Pearson Corr | T-test p-value | Levene p-value |
|---|---|---|---|---|---|---|---|
| | GPT-4 | Human | GPT-4 | Human | | | |
| Bard | 3.70 | 3.79 | 1.02 | 1.12 | 0.00 | 0.4078 | 0.4718 |
| ChatGPT | 5.00 | 4.42 | 0.83 | 0.97 | 0.00 | 0.0000 | 0.0001 |
| Claude2 | 4.69 | 3.91 | 0.70 | 1.12 | -0.00 | 0.0000 | 0.0000 |
| GPT4 | 5.17 | 4.77 | 0.71 | 0.93 | -0.00 | 0.0000 | 0.0001 |

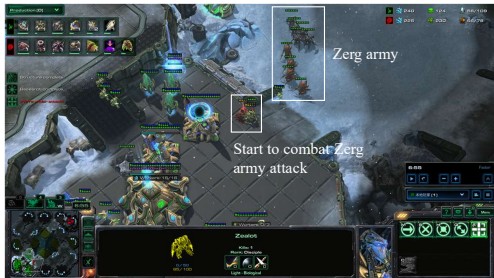

(a) Amidst rising tensions, the Zerg initiates an assault on the Protoss base.

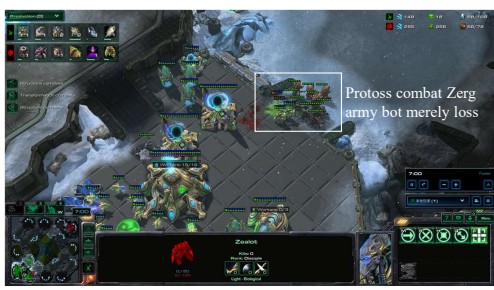

(b) The Zerg's roach and hydra assault gains momentum against the Protoss.

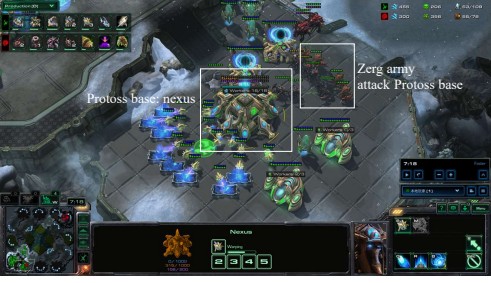

(c) The Zerg's onslaught continues, pressuring the Protoss's second base.

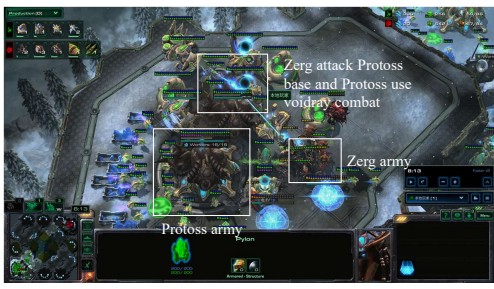

(d) The Zerg are attacking the Protoss main base, and the Protoss are defending with shield batteries and voidrays.

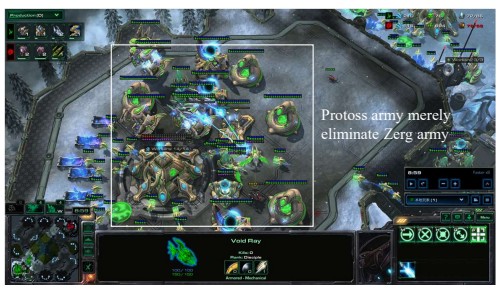

(e) Protoss defended with voidrays and phoenixes, leaving one roach damaging their weakened base.

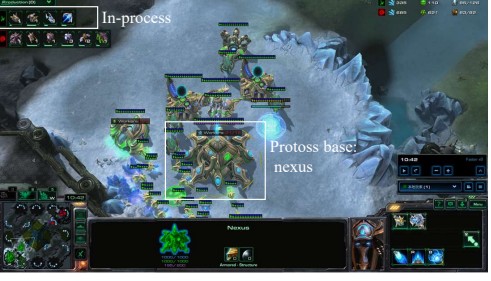

(f) Protoss prioritizes economy and voidray production, alongside researching phoenix range upgrade.

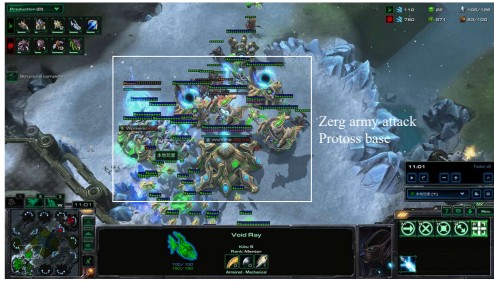

(g) Protoss defends Zerg's roach and hydra attacks with voidrays and phoenixes.

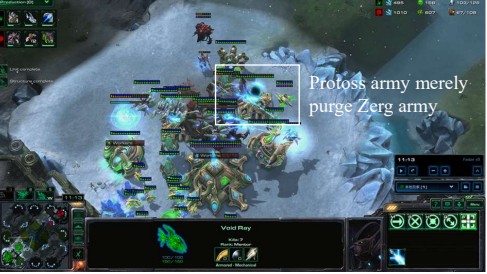

(h) Protoss protected their third base from Zerg roach and hydra assault with voidrays and phoenixes.

Figure 7: Protoss Strategic Defense Against Zerg Assaults: In the depicted scenes from StarCraft II, the LLM agent, controlling the Protoss, initially faces losses including the destruction of the second base during a Zerg Roach-Hydra assault. Despite this, the Protoss strategically expand their economy and transition to an air force with Void Rays and Phoenixes, successfully repelling a subsequent Zerg attack.

## C   Latency and Real-Time Feedback

The latency in our system is primarily determined by the inference speed and model size of the LLM. By leveraging fine-tuned open-source LLMs, our approach can achieve real-time feedback. Our *Chain of Summarization* (CoS) method is crucial in enabling this real-time feedback, significantly outperforming traditional CoT methods in terms of responsiveness.

For instance, with CoS, even smaller LLMs like the fine-tuned Qwen-2 7B can achieve real-time interaction (100 ms/step) and demonstrate competitive performance against human players. This level of real-time capability is not possible with standard CoT methods, highlighting the effectiveness of our CoS approach in facilitating swift strategic decision-making in complex, dynamic environments.

Although larger models such as GPT-4, Claude, and Gemini still face challenges in achieving real-time feedback (1s/step) due to their size, complexity, and network transmission factors, our CoS method significantly reduces their response times compared to CoT. This improvement allows for near-real-time performance in many scenarios. We are continuously optimizing our system to further enhance the real-time capabilities of these larger models.

Below are the delays observed for each step when different LLMs interact with the CoS method:

| Model Type | Model Name | Delay (each step) |
|---|---|---|
| Finetuned Open-Source LLM | Qwen2-7b | 98ms |
| Finetuned Open-Source LLM | Qwen2-1.8b | 64ms |
| Finetuned Open-Source LLM | LLAMA2-7B | 102ms |
| Closed-LLM | GPT3.5 | 1.2s |
| Closed-LLM | GPT4 | 2.3s |
| Closed-LLM | Gemini-PRO | 0.3s |

## D   Prompt and data Example

Figure 8, 9, 10, 11, and 12 depict sample data from the interaction process between the Chain of Summarization method and the TextStarCraft II environment.

There are two types of system prompts in the 7.1 section.

1. **Prompt1:** You are an AI trained in analyzing and summarizing StarCraft II games. You understand the nuances and strategies of the race race. Based on the summaries of multiple rounds in a game, we want you to analyze the game progression in a structured way. Your analysis should include the following aspects:

   (a) **Information Overview:** Provide a brief overview of the current situation based on all the rounds.
   (b) **Current Game Stage:** Determine the stage of the game based on the information of all rounds. Is it the early game, mid-game, or late game?
   (c) **Our Current Strategy:** From the information of all rounds, infer what our strategy might be.
   (d) **Enemy's Strategy:** Infer what the enemy's strategy might be, based on the information available.
   (e) **Key Information:** Highlight the most important aspects from all rounds that have significantly influenced the game.

2. **Prompt2:** You are an AI trained in analyzing and summarizing StarCraft II games. You understand the nuances and strategies of the self.race race. Based on the summaries of multiple rounds in a game, we want you to analyze the game progression in a structured way. Your analysis should include the following aspects:

   (a) **Game Overview:** Provide a brief overview of the current situation based on all the rounds.
   (b) **Current Game:** Determine the stage of the game based on the information of all rounds. Is it the early game, mid-game, or late game?

**1. Game time**
game_time: 15:27

**2. Resource information**
worker_supply: 47
mineral: 3005
gas: 4872
supply_left: 29
supply_cap: 131
supply_used: 102
base_count: 6
planning_base_count: 1

**3. Military information**
army_supply: 45
enemy_units_count: 33
enemy_UnitTypeId.INFESTOR: 2
enemy_UnitTypeId.ROACH: 16
enemy_UnitTypeId.HYDRALISK: 5

**5. Unit information**
Zealot_count: 0
stalker_count: 0
sentry_count: 0
adept_count: 0
high_templar_count: 0
dark_templar_count: 0
immortal_count: 0
colossus_count: 0
disruptor_count: 0
archon_count: 0
observer_count: 1
warp_prism_count: 0
phoenix_count: 0
voidray_count: 8
Oracle_count: 0
Carrier_count: 0
tempest_count: 0
mothership_count: 0

**8. Enemy Unit information**
enemy_UnitTypeId.INFESTOR: 2
enemy_UnitTypeId.ROACH: 16
enemy_UnitTypeId.HYDRALISK: 5
enemy_UnitTypeId.OVERSEER: 2
enemy_UnitTypeId.OVERLORDTRANSPORT: 3
enemy_UnitTypeId.ZERGLING: 3
enemy_UnitTypeId.QUEEN: 1
enemy_UnitTypeId.HYDRALISKBURROWED: 1

enemy_UnitTypeId.OVERSEER: 2
enemy_UnitTypeId.OVERLORDTRANSPORT: 3
enemy_UnitTypeId.ZERGLING: 3
enemy_UnitTypeId.QUEEN: 1
enemy_UnitTypeId.HYDRALISKBURROWED: 1

**4. Building information**
pylon_count: 7
gas_buildings_count: 10
gateway_count: 0
forge_count: 0
photon_cannon_count: 0
shield_battery_count: 4
warp_gate_count: 5
cybernetics_core_count: 1
twilight_council_count: 1
robotics_facility_count: 1
statgate_count: 4
templar_archives_count: 0

planning_worker_count: 0
planning_Zealot_count: 0
planning_stalker_count: 0
planning_sentry_count: 0
planning_adept_count: 0
planning_high_templar_count: 0
planning_dark_templar_count: 0
planning_immortal_count: 0
planning_colossus_count: 0
planning_disruptor_count: 0
planning_archon_count: 0.0
planning_observer_count: 0
planning_warp_prism_count: 0
planning_phoenix_count: 0
planning_voidray_count: 3
planning_Oracle_count: 0
planning_Carrier_count: 0
planning_tempest_count: 0
planning_mothership_count: 0

dark_shrine_count: 0
robotics_bay_count: 0
fleet_beacon_count: 0
planning_pylon_count: 0
planning_gas_buildings_count: 0
planning_gateway_count: 0
planning_forge_count: 0
planning_photon_cannon_count: 0
planning_shield_battery_count: 0
planning_warp_gate_count: 0
planning_cybernetics_core_count: 0
planning_twilight_council_count: 0
planning_robotics_facility_count: 0
planning_statgate_count: 1
planning_templar_archives_count: 0
planning_dark_shrine_count: 0
planning_robotics_bay_count: 0
planning_fleet_beacon_count: 0

**6. Research information**
warpgate_research_status: 1
protoss_air_armor_level_1_research_status: 1
protoss_air_armor_level_2_research_status: 0
protoss_air_armor_level_3_research_status: 0
protoss_ground_armor_level_1_research_status: 0
protoss_ground_armor_level_2_research_status: 0
protoss_ground_armor_level_3_research_status: 0
protoss_ground_weapon_level_1_research_status: 0
protoss_ground_weapon_level_2_research_status: 0
protoss_ground_weapon_level_3_research_status: 0
protoss_shield_level_1_research_status: 0
protoss_shield_level_2_research_status: 0
protoss_shield_level_3_research_status: 0

**7. Enemy Building information**
enemy_UnitTypeId.HATCHERY: 2
enemy_UnitTypeId.EXTRACTOR: 1
enemy_UnitTypeId.SPAWNINGPOOL: 1

Figure 8: An example of the TextStarCraft II's Raw text observation.

(c) **Our Situation:** Describe our current status in terms of:

    i. **Units and Buildings:** Analyze the state of our units and buildings.

    ii. **Economy:** Evaluate our economic condition, including resource collection and usage.

    iii. **Technology:** Describe the status of our technological research and what technologies we have unlocked so far. Analyze our technology tree, indicating the available and potential upgrades or units.

(d) **Our Strategy:** Infer our potential strategy based on our current situation and the information of all rounds.

(e) **Enemy's Strategy:** Infer the enemy's potential strategy, based on the available information.

(f) **Key Information:** Highlight the most important aspects from all rounds that have significantly influenced the game.

(g) **race specific prompt**

    i. Zerg :For Zerg, pay attention to whether there are enough larvae. If not, we should consider adding the INJECTLARVA command to the queue.

At 15:27 game time, our current StarCraft II situation is as follows:

**Resources:**
game_time: 15:27
worker_supply: 48
mineral: 3005
gas: 4872
supply_left: 33
supply_cap: 131
supply_used: 98
army_supply: 41
enemy_units_count: 32
base_count: 6

**Buildings:**
enemy_units_count: 32
base_count: 6
pylon_count: 7
gas_buildings_count: 10
shield_battery_count: 4
warp_gate_count: 5

cybernetics_core_count: 1
twilight_council_count: 1
robotics_facility_count: 1
statgate_count: 4
planning_base_count: 1
planning_statgate_count: 1

**Units:**
enemy_units_count: 32
base_count: 6
pylon_count: 7
gas_buildings_count: 10
shield_battery_count: 4
warp_gate_count: 5
cybernetics_core_count: 1
twilight_council_count: 1
robotics_facility_count: 1
statgate_count: 4
observer_count: 1
voidray_count: 7
planning_voidray_count: 3

**In-process:**
planning_base_count: 1
planning_statgate_count: 1
planning_voidray_count: 3

**Research:**
warpgate_research_status: 1
protoss_air_armor_level_1_research_status: 1

**Enemy Information:**
enemy_units_count: 32
enemy_UnitTypeId.ROACH: 15
enemy_UnitTypeId.OVERLORDTRANSPORT: 3
enemy_UnitTypeId.HYDRALISKBURROWED: 1
enemy_UnitTypeId.ZERGLING: 3
enemy_UnitTypeId.OVERSEER: 2
enemy_UnitTypeId.HYDRALISK: 5
enemy_UnitTypeId.INFESTOR: 2
enemy_UnitTypeId.QUEEN: 1
enemy_UnitTypeId.HATCHERY: 2
enemy_UnitTypeId.EXTRACTOR: 1
enemy_UnitTypeId.SPAWNINGPOOL: 1

Figure 9: L1 summarization

You are an AI trained in analyzing and summarizing StarCraft II games. You understand the nuances and strategies of the protoss (or zerg) race.

Based on the summaries of multiple rounds in a game, we want you to analyze the game progression in a structured way. Your analysis should include the following aspects:

1. Game Overview: Provide a brief overview of the current situation based on all the rounds.

2. Current Game Stage: Determine the stage of the game based on the information of all rounds. Is it the early game, mid-game, or late game?

3. Our Situation: Describe our current status in terms of:

    3.1 Units and Buildings: Analyze the state of our units and buildings.

    3.2 Economy: Evaluate our economic condition, including resource collection and usage.

    3.3 Technology: Describe the status of our technological research and what technologies we have unlocked so far. Analyze our technology tree, indicating the available and potential upgrades or units.

4. Our Strategy: Infer our potential strategy based on our current situation and the information of all rounds.

5. Enemy's Strategy: Infer the enemy's potential strategy, based on the available information.

6. Key Information: Highlight the most important aspects from all rounds that have significantly influenced the game.

For Protoss, keep an eye on Nexus's energy to Chrono Boost important structures.

Based on the game situation and strategies used by both sides, provide specific suggestions for the following areas:

1.Our Strategy: Propose adjustments to our current strategy to counter the enemy's moves and capitalize on our strengths.

2.Units and Buildings: Offer ways to enhance our unit composition and improve our building layout, suited to the current stage of the game.

3.Economy: Recommend better practices for resource gathering and usage, in line with our strategic needs.

4.Technology: Suggest focused research paths to gain technological advantages, considering our current research status and technology tree.

Lastly, consider the current situation and the suggestions provided, make {K} actionable and specific decisions from the action dictionary protoss_action_dict . This dictionary comprises four categories of actions: unit production, building construction, technology research, and other actions. Remember to align these decisions with the current stage of the game, and avoid proposing actions that are not currently feasible.

Figure 10: System prompt from L2 summrization. This prompt enables the Large Language Model to engage in playing StarCraft 2. The process includes several key steps: Situation Overview, Situation Analysis, Strategic Planning, Opponent Strategy Analysis, and Strategic Recommendations, concluding with the Decision-Making Process.

**1. Game Overview:**
At 15:28 game time, our situation is as follows: We have 49 workers, 2970 minerals, and 4884 gas. Our supply cap is at 131 with 33 supply left, and our army supply is 41. We have six bases and a decent number of buildings and units.

**2. Current Game Stage:**
Based on the game time and resource availability, we are still in the mid-game stage.

**3. Our Situation:**
  3.1 Units and Buildings: We have a diverse unit composition, including Voidrays, and a decent number of buildings, including a Cybernetics Core, Twilight Council, and Robotics Facility. We also have a good number of Pylons and gas buildings.
  3.2 Economy: Our economy is stable, with a sufficient number of workers and a good balance of minerals and gas.
  3.3 Technology: We have started researching Warpgate and Protoss Air Armor Level 1. However, there is room for further technological development and research.

**4. Our Strategy:**
Our strategy seems to be focused on air dominance with Voidrays. We should continue to leverage our air superiority while expanding our technological capabilities.

**5. Enemy's Strategy:**
 The enemy seems to be focused on a ground-based army composition, with Roaches, Zerglings, and Hydralisks. They also have some support units like Overseers and Infestors. We should be cautious of potential ground attacks and plan our defenses accordingly.

**6. Key Information:**
The most important aspect at this moment is our air superiority with Voidrays. However, we need to be prepared for potential ground-based attacks from the enemy's Roaches and Hydralisks. Our future success will depend on how we can maintain our air dominance while countering the enemy's ground forces.

Figure 11: LLM analysis: LLM take Situation Overview, Situation Analysis, Strategic Planning, Opponent Strategy Analysis these four stages.

    ii. Protoss:For Protoss, keep an eye on Nexus's energy to Chrono Boost important structures.

(h) Based on the game situation and strategies used by both sides, provide specific suggestions for the following areas:

    i. **Our Strategy**:Propose adjustments to our current strategy to counter the enemy's moves and capitalize on our strengths.

    ii. **Units and Buildings:**Offer ways to enhance our unit composition and improve our building layout, suited to the current stage of the game.

    iii. **Economy:**Recommend better practices for resource gathering and usage, in line with our strategic needs.

    iv. **Technology:**Suggest focused research paths to gain technological advantages, considering our current research status and technology tree.

(i) Lastly, consider the current situation and the suggestions provided, make K actionable and specific decisions from the action dictionary. This dictionary comprises four categories of actions: unit production, building construction, technology research, and other actions. Remember to align these decisions with the current stage of the game, and avoid proposing actions that are not currently feasible.

**LLM Suggestions & Decisions**

**Suggestions:**

**1.  Our Strategy:**
We should continue to produce Voidrays and maintain air superiority. However, we should also focus on developing a strong ground army to counter the enemy's Roaches and Hydralisks.

**2. Units and Buildings:**
Consider building more Gateways and producing units like Stalkers and Immortals to strengthen our ground army. Additionally, expanding our shield battery count can provide better defense against ground attacks.

**3. Economy:**
Ensure that our workers are efficiently gathering resources. Consider expanding to new bases to further boost our economy and support our unit production.

**4. Technology:**
Continue researching upgrades for our air units, such as Protoss Air Armor Level 2 and Level 3. Additionally, consider researching upgrades for our ground units, such as Protoss Ground Weapons and Armor.

**Decisions:**
0: <TRAIN STALKER>
1: <TRAIN IMMORTAL>
2: <BUILD GATEWAY>
3: <BUILD SHIELDBATTERY>
4: <EXPAND BASE>

Figure 12: LLM suggestions and decision: LLM give useful Strategic Recommendations and finally take the Decision-Making Process. Our Action-Extractor can extract the actions from Decisions. The actions of this infer are train stalker, train immortal, build gateway, build shield battery and expand base.

# E Alphastar and llm agent

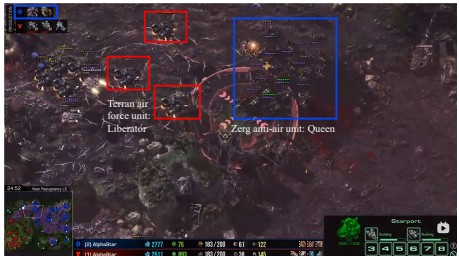

(a) Game time 24:52: Terran's Liberator-heavy army establishes control zones, while Zerg (AlphaStar) persists with Queens for defense, which is ineffective against the Liberators.

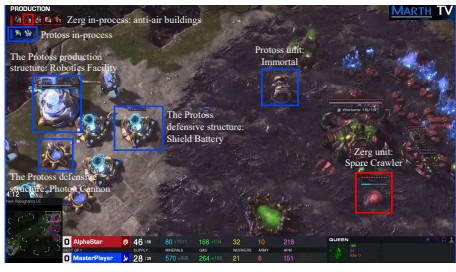

(b) Protoss (Master player) employs a Cannon, Shield Battery, and Immortal rush strategy. In response, Zerg (AlphaStar) builds Spore Crawlers for defense, which is ineffective against the current Protoss strategy.

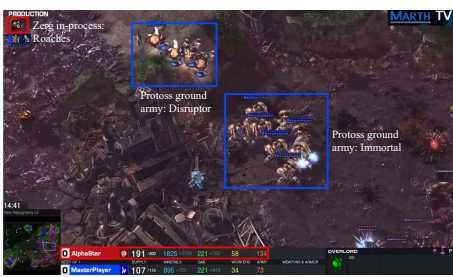

(c) Protoss (Master player) fields a heavy Immortal and Disruptor army. Despite a strong economy, Zerg (AlphaStar) continues producing Roaches, a unit countered by the Protoss forces, without opting for a transition.

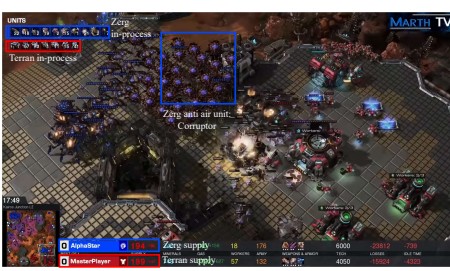

(d) Terran (Master player) employs a bio-tank composition (Marines, Medivacs, Marauders, Tanks) with minimal air units. Zerg (AlphaStar) counters with a large number of Corruptors, which are ineffective against Terran's ground-focused army.

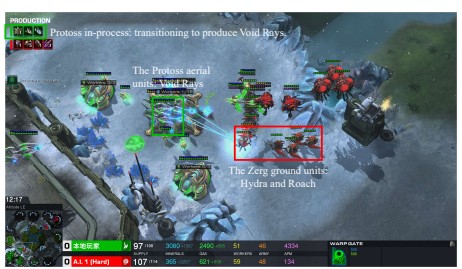

(e) Zerg (Built-in AI) utilizes a Hydra-Roach army composition, countered effectively by Protoss (LLM Agent) who defends with Voidrays and continues to produce more, capitalizing on their effectiveness against the Hydra-Roach combination.

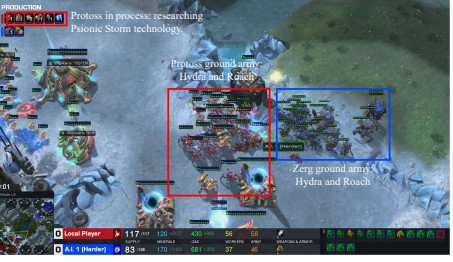

(f) Zerg (Built-in AI) employs a Hydra-Roach composition, countered by Protoss (LLM Agent) using Stalkers and Immortals for defense, while also researching Psionic Storm technology to effectively manage the Zerg forces.

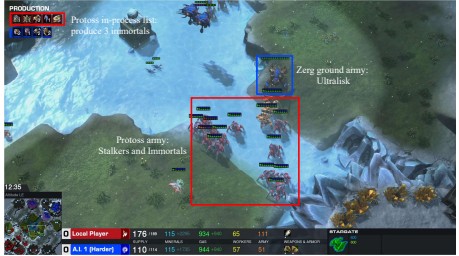

(g) Zerg (Built-in AI) fields an Ultralisk-heavy army, while Protoss (LLM Agent) responds with a composition of Stalkers and Immortals, actively producing more Immortals, highly effective against Ultralisks.

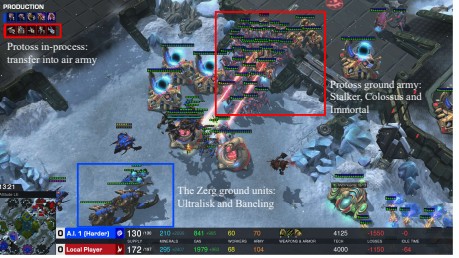

(h) Protoss (LLM agent) not only possesses units that counter the opponent's but also can expand its diverse array of strategies and army compositions, such as an air army.

Figure 13: The ability to adapt strategies based on the composition of enemy forces: AlphaStar (a, b, c, d) vs. LLM Agent (e, f, g, h).

# F Policy interpretability examples

In Section 6.3, we introduced the concept of policy interpretability within our LLM agent. We now present four illustrative examples to demonstrate how the LLM agent can generate reasonable strategies and decisions in TextStarCraft II. Each example comprises three components:

- *L1 Results Figure*: This displays the input of the multi-frame summarization method.
- *LLM Output*: The output of the multi-frame summarization method, which includes the LLM's analysis, suggestions, and decisions.
- *Decision Figure*: Illustrates the actual implementation within the StarCraft II game engine.

## F.1 Example 1

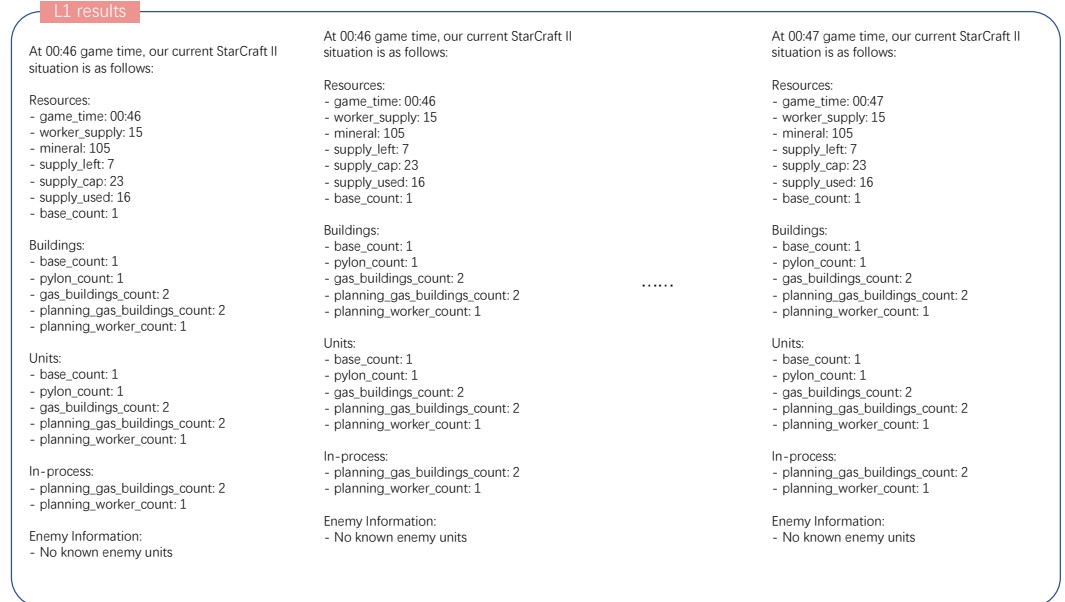

Figure 14: L1 results of 00:46 period

**LLM output:**

- 1. Game Overview: At 00:46 game time, our current situation is as follows. We have 15 workers and 105 minerals. Our supply cap is at 23 with 7 supply left, and our army supply is 16. We have one base, one pylon, and two gas buildings. There are no known enemy units.
- 2. Current Game Stage: Based on the limited resources and small army size, we are still in the early game stage.
- 3. Our Situation:
  - 3.1 Units and Buildings: We have a small worker force and limited infrastructure. We have one base, one pylon, and two gas buildings. There are no units present except for the workers.
  - 3.2 Economy: Our economy is just starting, with a small number of workers and limited resources.
  - 3.3 Technology: There is no evidence of any technological development or research at this stage.
- 4. Our Strategy: At this early stage, our strategy should focus on expanding our economy, establishing more bases, and increasing our worker count. **We should also consider scouting to gather information about the enemy's strategy.**

- 5. Enemy's Strategy: Since there is no information about enemy units, it is difficult to determine their strategy. However, it is important to remain vigilant and be prepared for any potential attacks.
- 6. Key Information: The most important aspect at this moment is to quickly expand our economy and increase our worker count. This will provide us with a stronger economic foundation and allow us to produce more units and buildings.

Suggestions:

- 1. Our Strategy: Prioritize expanding our economy by building additional bases and increasing our worker count. **Consider scouting to gather information about the enemy's strategy.**
- 2. Units and Buildings: Focus on building additional pylons to increase our supply cap and support future unit production. Consider constructing additional structures such as gateways or a robotics facility to start producing units.
- 3. Economy: Allocate workers to gather minerals and gas efficiently. Aim to saturate our current base and expand to new resource locations as soon as possible.
- 4. Technology: At this early stage, it is not necessary to focus on technology research. However, consider building a cybernetics core to unlock more advanced units and upgrades in the future.
- Decisions:
  - 0: <BUILD NEXUS>
  - 1: <BUILD PYLON>
  - 2: <BUILD GATEWAY>
  - 3: <TRAIN PROBE>
  - **4: <SCOUTING PROBE>**

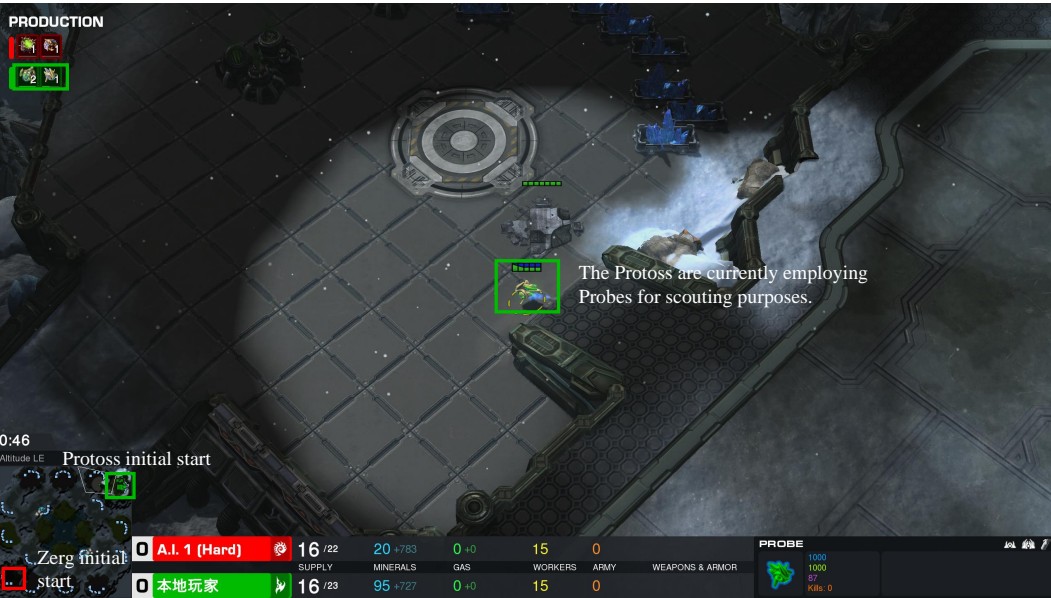

Figure 15: Decision of LLM Agent: Scouting probe

## F.2  Example 2

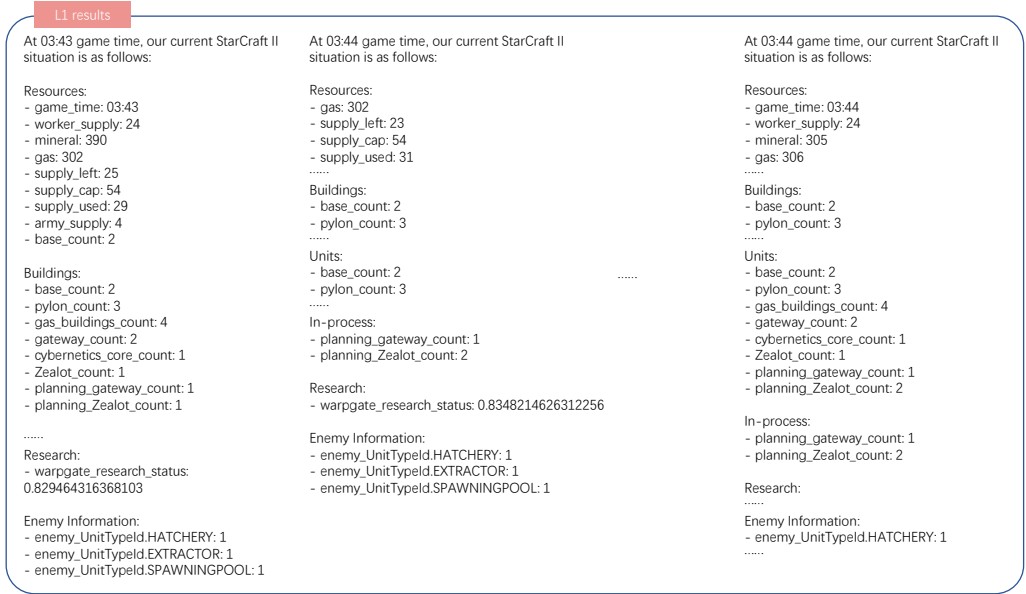

Figure 16: L1 results of 03:43 period

**LLM output:**

- 1. Game Overview: At 03:44 game time, our situation is relatively early in the game. We have 24 workers, 305 minerals, and 310 gas. Our supply cap is 54 with 23 supply left, and our army supply is 6. We currently have two bases.

- 2. Current Game Stage: Based on the resource levels and army size, we are still in the early game.

- 3. Our Situation:
  - 3.1 Units and Buildings: We have a limited number of units, with only 1 Zealot in our army. Our building count includes 3 pylons, 4 gas buildings, 2 gateways, and 1 Cybernetics Core. Additionally, we are planning to build 1 gateway and 2 Zealots.
  - 3.2 Economy: Our economy is modest, with a small worker supply and resource pool. It's crucial to boost our worker count and resource collection rate.
  - 3.3 Technology: **We have started researching Warpgate technology, which is a positive step.** However, our lack of advanced units indicates that our tech tree has not progressed significantly.

- 4. Our Strategy: Our current strategy appears to be a cautious early-game approach, focusing on tech development through Warpgate research.

- 5. Enemy's Strategy: The enemy seems to be expanding their Zerg infrastructure with a Hatchery, Extractor, and Spawning Pool. They might be preparing for Zergling production.

- 6. Key Information: The early game is a critical phase, and our limited unit count and resources indicate that we should prioritize both economy and army development. The ongoing Warpgate research suggests a transition to more advanced unit production.

Suggestions:

- 1. Our Strategy: **Continue with the Warpgate research as it will significantly improve our unit production capabilities.** Consider Chrono Boosting the Cybernetics Core to expedite this research.

- 2. Units and Buildings: Prioritize building more Zealots to bolster your army. Also, expand your Pylon network to increase your supply cap and ensure you don't get supply blocked.

- 3. Economy: Focus on worker production. Consider building more probes to saturate your mineral and gas collection, which will support both army and tech development.
- 4. Technology: **After completing Warpgate research**, evaluate your unit composition and decide which tech path to pursue based on the enemy's strategy. Consider building additional tech structures like a Twilight Council or a Robotics Facility.
- 5. Keep an eye on the enemy's expansion and unit production. Prepare for potential Zergling attacks by having Zealots ready for defense.
- Decisions:
  - 0: <RESEARCH WARPGATERESEARCH>
  - 1: <BUILD ZEALOT>
  - 2: <BUILD PROBE>
  - 3: <BUILD PYLON>
  - **4: <CHRONOBOOST CYBERNETICSCORE>**

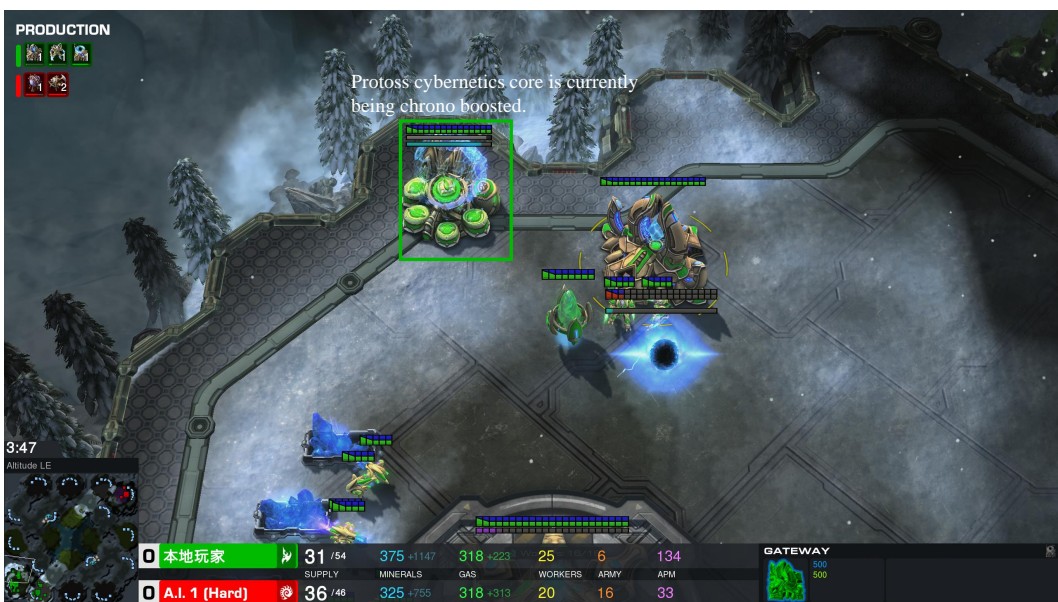

Figure 17: Decision of LLM Agent: Chronoboost Cybernetics Core

## F.3 Example 3

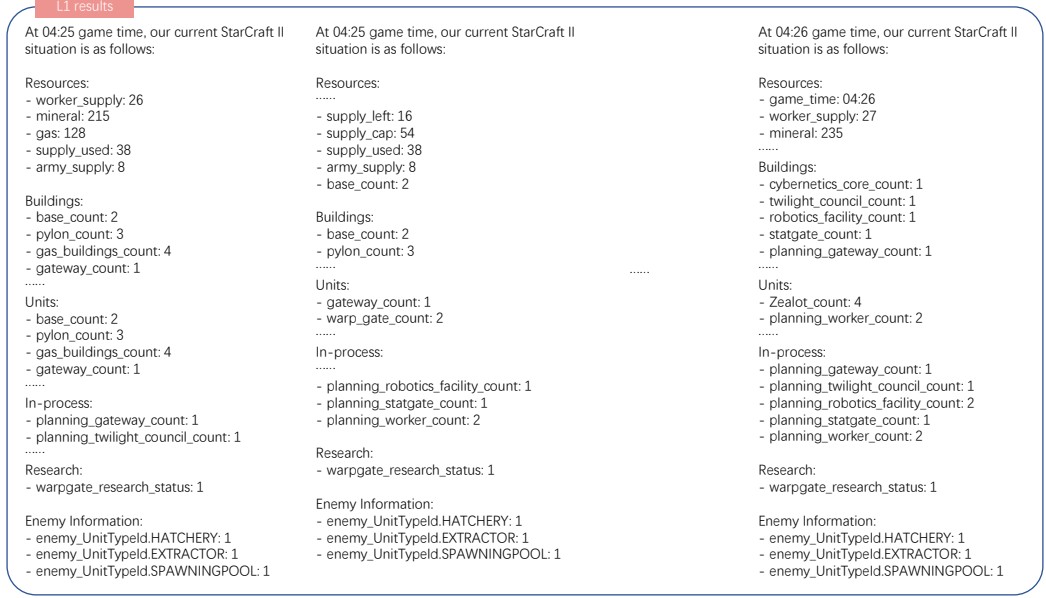

Figure 18: L1 results of 04:25 period

**LLM output:**

- 1. Game Overview: At 04:25 game time, our situation is quite challenging. We have limited resources with 215 minerals and 128 gas. Our supply cap is at 54 with 16 supply left, and our army supply is just 8. We currently have two bases.

- 2. Current Game Stage: Based on the limited resources and small army size, we are in the early game stage.

- 3. Our Situation:
  - 3.1 Units and Buildings: We have a modest army consisting of 4 Zealots. We are in the process of constructing additional Gateways, a Robotics Facility, and a Stargate.
  - 3.2 Economy: Our economy is struggling with minimal resources. We need to expand our resource collection urgently.
  - 3.3 Technology: We have already researched Warp Gate technology, which is a positive development. However, we lack diversity in our units and are still in the process of building advanced tech structures.

- 4. Our Strategy: It appears we are gearing up for a tech-heavy approach with the construction of advanced tech structures. However, our current economic state is not supporting this strategy well.

- 5. Enemy's Strategy: The enemy seems to be focusing on expanding with Hatchery and Extractor but has also started building a Spawning Pool, indicating the potential for Zerg unit production.

- 6. Key Information: The most crucial aspect at this moment is our resource scarcity. We need to address this issue immediately to support our tech-heavy strategy and build a more substantial army to defend against potential enemy threats.

Suggestions:

- 1. Our Strategy: Given our low resources, it's crucial to prioritize resource expansion in the short term. **Build additional Probes to maximize mineral and gas collection.** Consider a defensive strategy while we bolster our economy.

- 2. Units and Buildings: Continue constructing the planned tech structures, but also prioritize the construction of additional Gateways for faster unit production. Warp-in Zealots as a cost-effective defensive measure.

- 3. Economy: Expand to new resource locations to increase income. **Keep worker production consistent and utilize Chrono Boost on Nexus to accelerate Probe production.**

- 4. Technology: Complete the construction of the Robotics Facility and Stargate to unlock advanced unit options. Consider researching unit upgrades once these structures are operational. Given the early game situation, these adjustments will help us stabilize our economy and prepare for potential enemy threats.

- Decisions:
  - 0: <TRAIN PROBE>
  - 1: <BUILD GATEWAY>
  - 2: <EXPAND TO NEW RESOURCE LOCATION>
  - 3: <BUILD ROBOTICSFACILITY>
  - **4: <CHRONOBOOST NEXUS>**

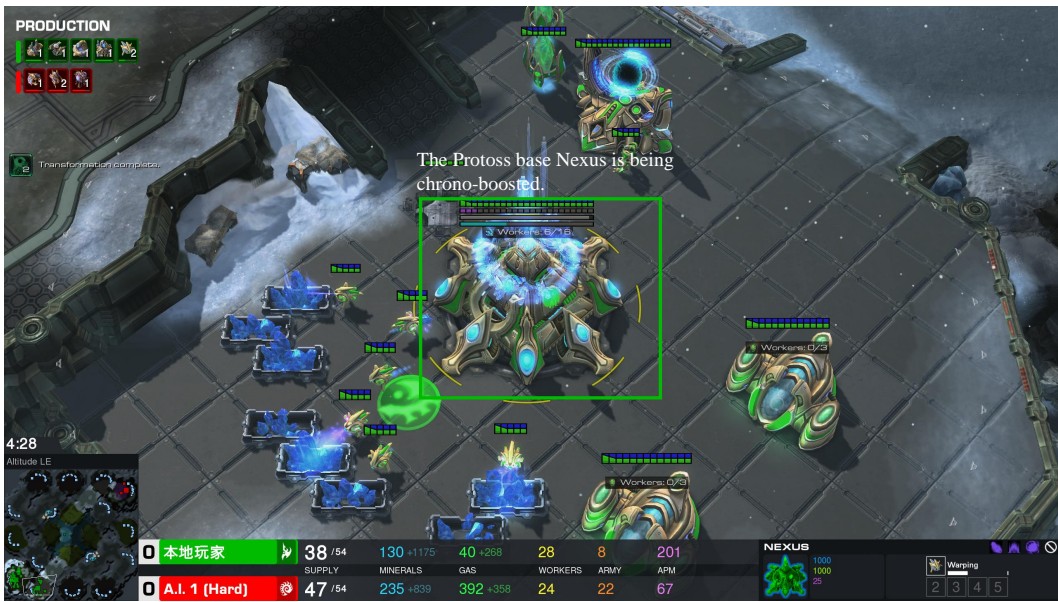

Figure 19: Decision of LLM Agent: Chronoboost Nexus

## F.4   Example 4

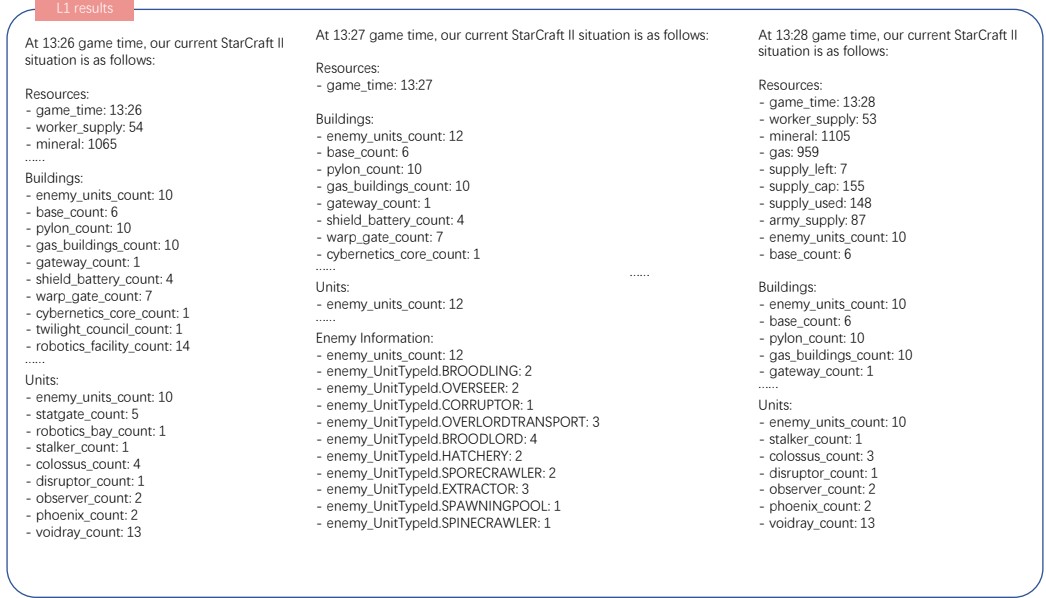

At 13:26 game time, our current StarCraft II situation is as follows:

Resources:
- game_time: 13:26
- worker_supply: 54
- mineral: 1065
......

Buildings:
- enemy_units_count: 10
- base_count: 6
- pylon_count: 10
- gas_buildings_count: 10
- gateway_count: 1
- shield_battery_count: 4
- warp_gate_count: 7
- cybernetics_core_count: 1
- twilight_council_count: 1
- robotics_facility_count: 14
......

Units:
- enemy_units_count: 10
- statgate_count: 5
- robotics_bay_count: 1
- stalker_count: 1
- colossus_count: 4
- disruptor_count: 1
- observer_count: 2
- phoenix_count: 2
- voidray_count: 13

At 13:27 game time, our current StarCraft II situation is as follows:

Resources:
- game_time: 13:27

Buildings:
- enemy_units_count: 12
- base_count: 6
- pylon_count: 10
- gas_buildings_count: 10
- gateway_count: 1
- shield_battery_count: 4
- warp_gate_count: 7
- cybernetics_core_count: 1
......

Units:
- enemy_units_count: 12
......

Enemy Information:
- enemy_units_count: 12
- enemy_UnitTypeId.BROODLING: 2
- enemy_UnitTypeId.OVERSEER: 2
- enemy_UnitTypeId.CORRUPTOR: 1
- enemy_UnitTypeId.OVERLORDTRANSPORT: 3
- enemy_UnitTypeId.BROODLORD: 4
- enemy_UnitTypeId.HATCHERY: 2
- enemy_UnitTypeId.SPORECRAWLER: 2
- enemy_UnitTypeId.EXTRACTOR: 3
- enemy_UnitTypeId.SPAWNINGPOOL: 1
- enemy_UnitTypeId.SPINECRAWLER: 1

At 13:28 game time, our current StarCraft II situation is as follows:

Resources:
- game_time: 13:28
- worker_supply: 53
- mineral: 1105
- gas: 959
- supply_left: 7
- supply_cap: 155
- supply_used: 148
- army_supply: 87
- enemy_units_count: 10
- base_count: 6

Buildings:
- enemy_units_count: 10
- base_count: 6
- pylon_count: 10
- gas_buildings_count: 10
- gateway_count: 1
......

Units:
- enemy_units_count: 10
- stalker_count: 1
- colossus_count: 3
- disruptor_count: 1
- observer_count: 2
- phoenix_count: 2
- voidray_count: 13

Figure 20: L1 results of 13:28 period

**LLM output:**

- 1. Game Overview: At 13:28 game time, our situation is as follows: We have 54 workers, 1135 minerals, and 975 gas. Our supply cap is at 155 with 7 supply left, and our army supply is 87. We have six bases.

- 2. Current Game Stage: Based on the game time and resource availability, we are in the mid to late-game stage.

- 3. Our Situation:
    - 3.1 Units and Buildings: We have a decent worker count and a strong army composition with Stalkers, Colossi, Disruptors, Observers, Phoenixes, and Void Rays. Our building count is also significant, with multiple pylons, gateways, robotics facilities, and a robotics bay.
    - 3.2 Economy: Our economy is stable, with a good balance of minerals and gas. We have enough workers to maintain our resource collection efficiently.
    - 3.3 Technology: We have already researched Warpgate technology, which is essential for unit production. However, there is no information about further technological advancements or upgrades.

- 4. Our Strategy: Our strategy seems to be focused on a strong ground and air army composition, utilizing Stalkers, Colossi, Disruptors, Phoenixes, and Void Rays. We should continue to leverage our army strength and maintain a strong economy.

- **5. Enemy's Strategy: The enemy has a diverse unit composition, including Broodlords, Overlord Transports, Corruptors, Overseers, and ground units. They seem to be focusing on air superiority and potentially using Broodlords for high ground control.**

- **6. Key Information: The most crucial aspect at this moment is our strong army composition, which includes anti-air units like Phoenixes and Void Rays. However, we should be cautious of the enemy's Broodlords, which can pose a significant threat to our ground forces. Additionally, we need to focus on further technological advancements and upgrades to stay ahead in the game.**
  Suggestions:

– **1. Our Strategy: We should continue to strengthen our army composition by adding more anti-air units like Phoenixes and Void Rays. Additionally, we should consider countering the enemy's Broodlords with appropriate units or tactics.**

– **2. Units and Buildings: Given the enemy's air dominance, it would be beneficial to increase the production of Phoenixes and Void Rays. Additionally, consider building more Stargates to support the production of these units. It may also be necessary to reinforce our ground forces with additional Stalkers and Colossi.**

– 3. Economy: Maintain a steady worker production and ensure efficient resource gathering. Consider expanding to new resource locations to support the production of a larger army.

– 4. Technology: Research upgrades that enhance the effectiveness of our units, such as air weapon and armor upgrades for Phoenixes and Void Rays. Additionally, consider researching extended thermal lance for increased Colossus range.

Decisions:

– **0: <TRAIN PHOENIX>**
– **1: <TRAIN VOIDRAY>**
– **2: <BUILD STARGATE>**
– 3: <TRAIN STALKER>
– **4: <TRAIN COLOSSUS>**

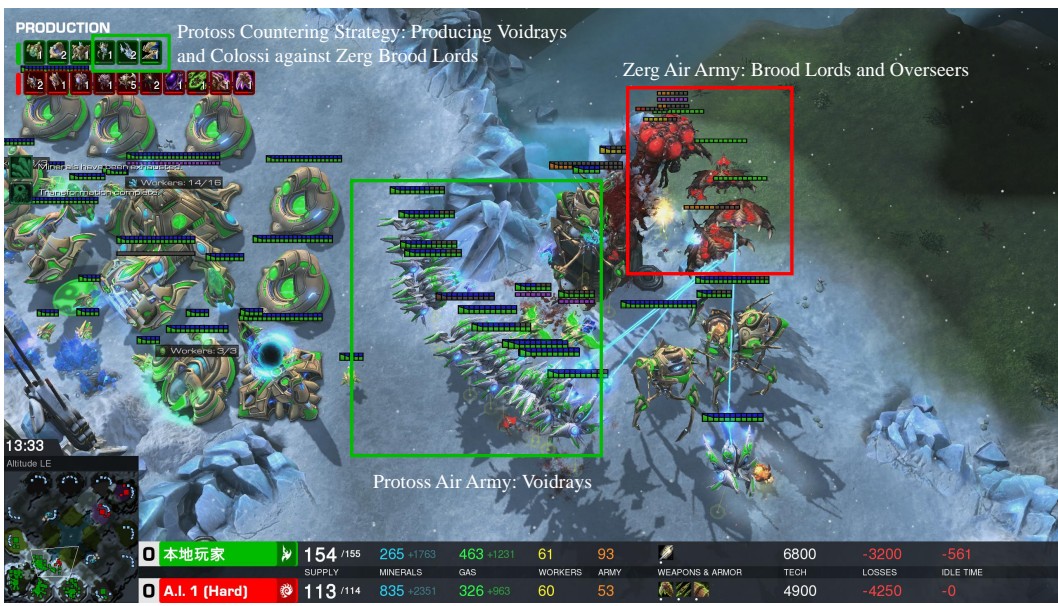

Figure 21: Decision of LLM Agent: Adapt strategy in response to opponent's tactics.

# G  Questions and LLM Responses

In subsection 6.2, we documented the responses from various LLMs, encompassing GPT3.5, GPT4, Claude2, and Bard, spanning five distinct question categories.

1. Q1:Basic Knowledge of StarCraft2

   (a) Are you familiar with StarCraft2?
   (b) Could you provide an overview of StarCraft2,including its races, esports scene,and other pertinent details?

2. Q2:Racial Mechanics in StarCraft2 Discuss the unique mechanisms of the following races:Zerg,Protoss and Terran.

3. Q3:Typical Strategies in StarCraft2 for Different matchups?

   (a) TVZ:What's a common strategy for Teeran vs Zerg?
   (b) TVT:What's a typical srategy in Terran vs Terran matchups?
   (c) TVP:How do Terran players generally approach a Protoss opponent?
   (d) ZVZ:What's standard strategy in Zerg vs Zerg battles?
   (e) ZVP:What are Zerg strategies when facing Protoss?
   (f) ZVT:How do Zerg Players counter Terran tactis?
   (g) PVZ:How should Protoss approach a Zerg opponent?
   (h) PVP:What's the strategy for Protoss vs Protoss gams?
   (i) PVT:How do Protoss players generally strategize against Terran?

4. Q4:Ability to Outline Build Orders for Different Races. Can you provide a standard build order for each race?

5. Q5: Classic Strategies and Counterplays in StarCraft2

   (a) PVZ:What are effective strategies and counterplays against a roach-hydra attack?
   (b) TVZ:How should Terran approach a mutalisk-zergling-baneling composition?Are there gambits or counter-strategies to consider?
   (c) TVP:When faced with an archon-zealot push,what are the bast strategies and counters for Terran?

## G.1  Basic Knowledge of StarCraft II

**Q1: Are you familiar with StarCraft II?**

**GPT3.5:**  Yes, I'm familiar with StarCraft II. StarCraft II is a popular real-time strategy (RTS) video game developed and published by Blizzard Entertainment. It is the sequel to the original Star-Craft and was released in three separate episodes: Wings of Liberty, Heart of the Swarm, and Legacy of the Void. The game is set in a science fiction universe and features three distinct factions: the Terrans, the Zerg, and the Protoss, each with its own unique units and abilities. Players in StarCraft II engage in strategic and tactical battles, gathering resources, building armies, and competing against other players or AI opponents. The game has a strong competitive scene, with professional players and tournaments, and it has a dedicated fanbase worldwide. If you have any specific questions about StarCraft II or need information on a particular aspect of the game, feel free to ask!

**GPT4:**  Yes, I'm familiar with "StarCraft II," the popular real-time strategy (RTS) game developed and published by Blizzard Entertainment. The game is a sequel to the original "StarCraft" from 1998 and is set in a distant part of the Milky Way galaxy in the 26th century. It maintains the classic RTS elements of gathering resources, building bases, and commanding armies to defeat opponents. "StarCraft II" consists of three main installments representing each of the game's species: (1) "Wings of Liberty" (released in 2010) - This is the base game where the primary campaign focuses on the Terrans, specifically on the character Jim Raynor, who is leading a rebellion against the autocratic Terran Dominion. (2) "Heart of the Swarm" (released in 2013) - This expansion focuses on the Zerg race and follows the story of Sarah Kerrigan (formerly a Terran who was transformed into the Queen of Blades, the leader of the Zerg) in her quest to reunite the Zerg Swarm and exact revenge on the

Terran Dominion's emperor, Arcturus Mengsk. (3) "Legacy of the Void" (released in 2015) - The final installment concentrates on the Protoss and features the character Artanis as he works to unite the Protoss and confront a cosmic threat known as the "Dark Voice" or "Amon."

The game is recognized for its competitive multiplayer mode, where players from around the world compete in leagues. Each of the three races in the game (Terrans, Zerg, and Protoss) has its unique set of units and structures, which results in very different strategies and requires a deep understanding of the game to master."StarCraft II" has also been a significant part of the professional e-sports scene, with numerous tournaments worldwide, including the prestigious StarCraft II World Championship Series (WCS). The game's emphasis on quick decision-making, strategic diversity, and real-time tactical execution contribute to its standing as a respected e-sport.

**Claude2:**    Yes, I'm familiar with StarCraft 2! It's a popular real-time strategy video game developed by Blizzard Entertainment. Here are some key facts about StarCraft 2:

- It's the sequel to the original StarCraft game that came out in 1998. StarCraft 2 was released in 2010.

- There are three installments or chapters in StarCraft 2 - Wings of Liberty, Heart of the Swarm, and Legacy of the Void.

- Players can play as three factions or races - the Terrans (humans), the insect-like Zerg, and the technologically advanced Protoss.

- The gameplay involves gathering resources, building bases and armies, and battling opponents in real-time. The goal is to destroy the enemy base.

- StarCraft 2 has a thriving eSports scene with many professional tournaments and leagues. Top players can earn significant prize money.

- It's considered one of the greatest and most influential real-time strategy franchises to date, known for its balanced asynchronous gameplay, strategic depth, and cinematic storytelling.

- Blizzard continues to support StarCraft 2 with updates, new co-op commanders, and balance changes to keep the multiplayer aspect fresh.

Let me know if you have any other specific questions about StarCraft 2! I'm happy to discuss strategies, lore, competitive play, or anything else related to this classic RTS game.

**Bard:**    Yes, I am familiar with StarCraft II. It is a science fiction real-time strategy video game developed and published by Blizzard Entertainment. It was released worldwide in July 2010 for Microsoft Windows and Mac OS X. StarCraft II is the sequel to the 1998 game StarCraft, and it is set four years after the events of the original game. StarCraft II features three playable races: the Terrans, the Zerg, and the Protoss. Each race has its own unique units and abilities, and the player must use their strengths to defeat their opponents. StarCraft II can be played in single-player mode, cooperative mode, or multiplayer mode. The single-player campaign in StarCraft II is divided into three parts: Wings of Liberty, Heart of the Swarm, and Legacy of the Void. Each campaign follows the story of a different character from one of the three races. The cooperative mode in StarCraft II allows players to team up with other players to complete missions against computer-controlled enemies. The multiplayer mode in StarCraft II allows players to compete against each other in ranked or unranked matches. StarCraft II is a very popular game, and it has a large and dedicated esports scene. StarCraft II tournaments are held all over the world, and some of the best players in the world can earn millions of dollars in prize money.

Is there anything else you would like to know about StarCraft II?

**Q2: Could you provide an overview of StarCraft II, including its races, esports scene, and other pertinent details?**

**GPT3.5:**    Certainly! Here's an overview of StarCraft II, including its races, esports scene, and other important details:

Game Overview:

- Release: StarCraft II was released in three separate episodes. The first episode, "Wings of Liberty," was released in July 2010. The second episode, "Heart of the Swarm," followed in March 2013, and the final episode, "Legacy of the Void," was released in November 2015.
- Genre: StarCraft II is a real-time strategy (RTS) game that focuses on resource management, base building, and strategic combat.
- Developer and Publisher: Developed and published by Blizzard Entertainment.
- Setting: The game is set in a futuristic science fiction universe and features three distinct races: Terrans, Zerg, and Protoss. Each race has its own unique units, buildings, and gameplay mechanics.
  Single Player: Each episode of StarCraft II has a single-player campaign that follows the story of one of the three races. The campaigns offer engaging storytelling and missions with varying objectives.

Races:

- Terrans: Terrans are humans who have been exiled from Earth and are known for their adaptability and use of advanced technology. They have a wide range of mechanical units and structures.
- Zerg: Zerg are a highly adaptable and insectoid alien race that can rapidly grow and overwhelm their opponents with numbers. They are known for their biological units and ability to spread creep.
- Protoss: Protoss are a technologically advanced alien race with powerful psionic abilities. They have advanced units and structures, and their gameplay is focused on precision and strategy.

Esports Scene:

StarCraft II has a thriving esports scene, with numerous tournaments and professional players from around the world.Major tournaments include the World Championship Series (WCS), Intel Extreme Masters (IEM), GSL (Global StarCraft II League), and BlizzCon.Professional players and teams compete for significant prize money and recognition in these tournaments.The game's competitive nature and high skill ceiling make it a popular choice for both casual and professional players.

Legacy and Impact:

StarCraft II is the successor to the original StarCraft, which is considered one of the most influential and iconic real-time strategy games of all time.StarCraft II continued the legacy of its predecessor and expanded the game's lore and gameplay mechanics.The game has a dedicated fanbase and has had a significant impact on the esports industry.

Community and Custom Content:

StarCraft II has a vibrant modding and custom content community. Players can create and share custom maps, game modes, and mods through the StarCraft II Arcade.This community-driven aspect adds to the game's longevity and diversity of gameplay experiences.While my knowledge is based on information available up until September 2021, StarCraft II has likely continued to evolve and develop in the years since. If you have specific questions or need more recent information about the game or its esports scene, I recommend checking official sources and community forums for the latest updates.

**GPT4:** "StarCraft II" is a monumental figure in the world of real-time strategy (RTS) games and e-sports. Developed by Blizzard Entertainment, it's known for its demanding skill level, strategic diversity, and an active professional scene. Here's a comprehensive overview:

1. Overview:

Released in 2010, "StarCraft II" is set in a science fiction universe in the 26th century and focuses on the interstellar struggles between three species: the Terrans (humans exiled from Earth), the Zerg (a super-species of assimilated life forms), and the Protoss (a technologically advanced species with psionic abilities).

2. Gameplay:

Players choose one of three races to control, each offering a unique array of units and technologies. Games typically involve gathering resources, expanding bases, and commanding armies to defeat enemies. The game's asymmetrical balance is a defining characteristic, with each race having different units, strengths, and weaknesses, requiring distinct strategies.

3. Races:

Terrans: Known for their adaptability and defensive capabilities, Terrans have versatile units and a unique building mechanic (buildings can lift off and move). They are human exiles from Earth. Zerg: The Zerg are all about rapid proliferation and adaptation. They produce large numbers of units quickly and can evolve them into more advanced forms. They rely on overwhelming their opponents with sheer numbers and have the ability to rapidly spread across the map. Protoss: Protoss units and buildings are generally more expensive (in terms of in-game resources) but highly powerful and efficient. They have strong shields, advanced technology, and can warp units directly onto the battlefield.

4. Campaigns:

"StarCraft II" was released as a trilogy, with each installment focusing on a different race's perspective in the ongoing interstellar war. "Wings of Liberty": Focuses on the Terran campaign. "Heart of the Swarm": Revolves around the Zerg campaign. "Legacy of the Void": Concludes with the Protoss campaign.

5. E-Sports:

Shortly after its release, "StarCraft II" exploded into the e-sports scene. It built on the legacy of the original "StarCraft" game, which was one of the first globally recognized e-sports. The game is celebrated for its incredibly high skill ceiling and strategic depth, making professional matches fascinating for spectators. The "StarCraft II World Championship Series (WCS)" was the premier competitive circuit for professional "StarCraft II" players, featuring several regional events leading up to a global year-end championship. South Korea, known for pioneering e-sports, has been particularly prominent in the "StarCraft II" scene, producing numerous professional players who dominate in international competitions.

6. Community and Legacy:

"StarCraft II" has a strong community of players, fans, and dedicated content creators. The game's custom map editor has led to a variety of player-created content, adding to the game's longevity. In October 2020, Blizzard announced the end of active development of new content for "StarCraft II," shifting to a maintenance mode for the game with a focus on balance and ecosystem health. Despite the reduction in developer-driven content, the community and e-sports scene remain active, with third-party organizers continuing to hold tournaments.

7. Free-to-Play Transition:

In 2017, "StarCraft II" became free-to-play, allowing players to access the original "Wings of Liberty" campaign and the multiplayer modes for free, which helped maintain the game's player base. The legacy of "StarCraft II" in the gaming community is marked by its technical and strategic depth, the skill it demands from its players, and its significant impact on the broader e-sports industry.

**Claude2:** Absolutely, here's an overview of StarCraft II:

- Races: There are three playable races in StarCraft II - the Terrans (humans), the insectoid Zerg, and the technologically advanced Protoss. Each race has distinct units, structures, abilities, and playstyles.

- Terrans are versatile and mobile, with units like Marines, Tanks, and Battlecruisers. They can adapt quickly and have a lot of defensive and zone control options.

- Zerg are focused on overwhelm and expansion, capable of producing large armies rapidly. Key units are Zerglings, Roaches, Hydralisks and Mutalisks. They can overrun opponents through sheer numbers.

- Protoss have advanced technology and psionic abilities. Their units like Stalkers, Immortals and Carriers are slower but more powerful. They rely on coordination and positioning in engagements.

- eSports Scene: StarCraft II has a longstanding professional esports scene in Korea especially. Top players compete in premier tournaments like GSL and Proleague for prize money and glory.

- Popular pro players include Serral, Maru, Rogue, Dark and soO. Blizzard directly supports the scene through the ESL Pro Tour and other circuits.

- Game Modes: The main competitive mode is 1v1, but 2v2, 3v3 and 4v4 team games are also popular. There is also a Co-op PVE mode where players team up against AI enemies. Custom games allow for all kinds of player-created variants.

- Expansions: Wings of Liberty, Heart of the Swarm and Legacy of the Void are the three installments that comprise StarCraft II. Each focuses on one race's campaign story while expanding multiplayer.

Let me know if you need any clarification or have additional questions!

**Bard:** Sure. Here is an overview of StarCraft II, including its races, esports scene, and other pertinent details:

Races:StarCraft II features three playable races: the Terrans, the Zerg, and the Protoss. Each race has its own unique units and abilities, and the player must use their strengths to defeat their opponents.

- Terrans: The Terrans are a versatile race that combines human ingenuity with advanced technology. They have a wide variety of units to choose from, including Marines, Siege Tanks, and Battlecruisers. Terrans are also known for their powerful mobile structures, such as Command Centers and Factories.

- Zerg: The Zerg are a swarm-based race that overwhelms their enemies with sheer numbers. They have a variety of cheap and expendable units, such as Zerglings and Roaches, as well as more powerful units such as Hydralisks and Ultralisks. Zerg players are also known for their ability to quickly expand their bases and produce units.

- Protoss: The Protoss are a highly advanced race that relies on powerful technology and psionic abilities. They have a limited number of units to choose from, but each unit is very powerful. Protoss players are known for their strong economies and their ability to build powerful armies.

Esports scene

StarCraft II has a large and dedicated esports scene. StarCraft II tournaments are held all over the world, and some of the best players in the world can earn millions of dollars in prize money.The biggest StarCraft II tournament of the year is the Global StarCraft II League (GSL). The GSL is a premier tournament series that features the best players from all over the world. Other major StarCraft II tournaments include the Intel Extreme Masters (IEM) and the World Championship Series (WCS).

Other pertinent details

- StarCraft II is a very challenging game, and it takes a lot of practice to master. The game is also very popular among competitive gamers, and it has a very active esports scene.Here are some other pertinent details about StarCraft II:

- StarCraft II is a free-to-play game. The base game includes the single-player campaign for Wings of Liberty, as well as the multiplayer mode. The other two campaigns, Heart of the Swarm and Legacy of the Void, can be purchased separately.

- StarCraft II has a very active modding community. There are thousands of mods available for StarCraft II, which can add new units, maps, and gameplay mechanics to the game.

- StarCraft II is a very popular game among streamers. There are many professional StarCraft II streamers who broadcast their gameplay to viewers on Twitch and YouTube.

If you are interested in learning more about StarCraft II, I recommend visiting the official Blizzard website or watching some StarCraft II streams on Twitch or YouTube.

## G.2    Racial Mechanics in StarCraft II

**Q: Discuss the unique mechanisms of the following races: Zerg, protoss and Terran**

**GPT3.5:** Certainly! Each of the three races in StarCraft II has its own unique mechanics and gameplay style. Here's an overview of the racial mechanics for the Zerg, Protoss, and Terran factions:

Zerg:

- Creep Spread: Zerg players can spread "creep" across the map. Creep is a slimy substance that not only provides vision but also enhances the movement speed of Zerg units when they are on it. Creep is generated by buildings called "Creep Tumors" and the primary Zerg building, the Hatchery or Lair.

- Larva Mechanic: Zerg units are produced from Larvae, which are spawned by Hatcheries and Lairs. The number of Larvae determines how quickly a Zerg player can build units. Managing Larvae effectively is crucial to Zerg gameplay.

- Biological Units: The Zerg rely heavily on biological units and are known for their ability to quickly swarm opponents with large numbers of units. Their units often have regenerative abilities and can be upgraded to various forms. Evolution Chambers: Zerg units can be upgraded at Evolution Chambers, allowing players to adapt their units to counter their opponent's strategies.

Protoss:

- Warp-In Mechanic: Protoss units can warp in directly onto the battlefield from Pylons or Warp Gates, providing a unique mobility advantage. However, this requires strategic placement of Pylons and Warp Gates.

- Shields: Many Protoss units have energy-based shields in addition to their regular health. Shields regenerate over time, making hit-and-run tactics effective when used wisely.

- Psionic Abilities: Protoss units often have psionic abilities that can be used for offense or defense. Examples include Force Fields, which can block enemy units, and Psionic Storm, which deals area damage.

- Warpgate Research: Protoss players can research the "Warpgate" technology, allowing them to transform their Gateways into Warp Gates, significantly increasing their flexibility in producing units.

Terran:

- MULEs and SCVs: Terran players can call down MULEs (Mechanized Utility Drones) to quickly gather resources. SCVs (Space Construction Vehicles) can also repair buildings and mechanical units.

- Building Mobility: Terran buildings are often mobile, with the ability to lift off and land in new locations. This flexibility allows for repositioning and strategic play.

- Supply Depots and Wall-offs: Terran players use Supply Depots to provide supply and can use them to create wall-offs to block enemy movement.

- Terran Infantry: Terran infantry units can be upgraded with a variety of weapons and equipment, making them versatile and adaptable to different situations.

- Siege Tanks: Terran players often employ Siege Tanks, which can transform into siege mode to deal massive area damage. This unit is a cornerstone of Terran defensive strategies.

Each race in StarCraft II offers a distinct and challenging gameplay experience, and mastering their unique mechanics is essential for success in the game. Players often choose a race that suits their preferred playstyle, whether it's the swarm tactics of the Zerg, the advanced technology of the Protoss, or the adaptability of the Terran.

**GPT4:** "StarCraft II" stands out in the RTS genre due to its distinct, asymmetric gameplay, meaning each race has unique mechanisms, units, and strategies that create a deep, strategic environment. Here's a detailed look at the specific racial mechanics for the Zerg, Protoss, and Terran factions:

1. Zerg:

- Creep: One of the most distinctive aspects of the Zerg is their reliance on Creep, a living mat of bio-organic substance that spreads from their main structures. Zerg buildings, except for the Hatchery, must be built on Creep. It also allows Zerg units to move faster when they are on it, providing both strategic and defensive advantages.
- Larvae and Macro Mechanics: Zergs uniquely produce units through Larvae generated by their primary structure, the Hatchery (which can be upgraded to a Lair and then a Hive for more advanced units). These Larvae morph into different units, meaning the Zerg can adapt their production in real-time as they see fit. The Queen unit can also spawn additional Larvae to speed up production, emphasizing the Zerg's swarm-oriented strategy.
- Rapid Expansion: Zerg players often rapidly colonize areas of the map to gain more resources, using inexpensive units to overwhelm opponents. Their ability to quickly spread Creep helps in establishing new bases and fortifying territories.

2. Protoss:

- Warp Gate Technology: A definitive feature for the Protoss is their advanced technology, represented through the transformation of their Gateway buildings into Warp Gates. This allows the production and immediate deployment of units across the battlefield, provided there is a power field (generated by structures like the Pylon). This ability is crucial for both reinforcing units during a battle and executing surprise attacks.
- Shields and Health: Protoss units and buildings have two layers of health: a regenerative shield and their actual health. This characteristic allows Protoss armies to be very resilient, as shield points regenerate over time when not under attack.
- Chrono Boost: The Protoss Nexus, the main base structure, has an ability called Chrono Boost that can be used to speed up the production of units, research, and buildings. This feature is integral to the race's adaptability, allowing them to respond quickly to threats.

3. Terran:

- Building Mobility: Terran structures, unique amongst the races, have the ability to lift off and relocate, allowing for strategic repositioning, surprise tactics, and base migration. This feature is particularly useful for avoiding threats, scouting, and creating forward unit production points.
- MULEs and Macro Mechanics: Terran Command Centers can call down MULEs (Mining Unit, Lifting Equipment), temporary units that gather resources at a much higher rate than SCVs, the standard worker unit. This ability provides a significant boost to the Terran economy over short periods, allowing for bursts of increased production.
- Bunkers and Siege Warfare: The Terrans have a defensive style of play, with structures like Bunkers and units like the Siege Tank. Bunkers can be garrisoned with infantry units to improve their survivability and increase the Bunker's firepower. Siege Tanks have a transformation ability, allowing them to switch to a stationary siege mode with increased range and damage, ideal for both defense and area control.

Each race's unique mechanics require players to adopt very different strategies for each, contributing to the game's dynamic competitive environment. Mastering the utilization of these racial featuresalong with understanding those of potential opponentsis crucial for success in "StarCraft II."

**Claude2:** Here's an overview of the unique racial mechanics for each of the three StarCraft II races:

Zerg

- - Larva - Zerg units are produced from larva generated by Hatcheries. This allows the Zerg player to quickly amass large armies.
- - Creep - Zerg buildings must be built on creep, the purple biological carpet produced by Hatcheries and Creep Tumors. Units move faster on creep.
- - Inject Larva - Queens can inject larva into Hatcheries to boost unit production. Proper injecting is essential for Zerg macro.

- - Morphing - Many Zerg units can morph or evolve into more powerful variants. Examples: Banelings, Lurkers, Brood Lords.

Protoss

- - Warp-in - Protoss can warp-in units directly onto the battlefield provided they have a Pylon power source and Gateway/Warp Gate ready. Critical for reinforcements.
- - Chrono Boost - The Nexus's Chrono Boost ability accelerates building and research times. Important for protoss economy and tech.
- - Psionics - Many protoss units have psionic abilities like force fields, hallucination and psi storms that provide battlefield control.
- - Power Fields - Pylons power protoss buildings and enable warp-in. Destroying Pylons cuts power and supply for an area.

Terran

- - MULEs - SCVs can build MULEs at Orbital Commands that harvest resources at a faster rate than SCVs. Boosts income.
- - Salvage - Most Terran structures can be salvaged to recoup some of the resources invested into them. Provides economic flexibility.
- - Add-ons - Buildings like Barracks and Factories can have Reactor or Tech Lab add-ons to enhance unit production capabilities.
- - Transforming Units - Many Terran units like Hellions and Vikings can transform between modes to adapt as needed during battle.

Let me know if you need any clarification or have additional questions!

**Bard:** Zerg

The Zerg are a swarm-based race that overwhelms their enemies with sheer numbers. They have a variety of cheap and expendable units, such as Zerglings and Roaches, as well as more powerful units such as Hydralisks and Ultralisks. Zerg players are also known for their ability to quickly expand their bases and produce units.

Here are some of the unique racial mechanics of the Zerg:

- Inject Larva: Zerg Hatcheries produce Larva, which can then be morphed into different types of units. The Inject Larva ability allows Zerg players to increase the production rate of their Hatcheries.
- Creep Spread: Zerg Creep is a substance that spreads from Zerg structures and units. Zerg units gain movement speed and attack speed on Creep. Zerg players can use Creep to quickly move their armies around the map and to gain an advantage in battles.
- Overlord Speed: Overlords are Zerg flying units that provide vision and can transport units. Zerg players can upgrade the speed of their Overlords, which allows them to quickly scout the map and to transport their armies to different locations.
- Zerg Rush: A Zerg Rush is a strategy where Zerg players produce a large number of cheap and expendable units early in the game and attack their opponent. Zerg Rushes can be very effective, but they can also be countered by players who are prepared for them.

Protoss

The Protoss are a highly advanced race that relies on powerful technology and psionic abilities. They have a limited number of units to choose from, but each unit is very powerful. Protoss players are known for their strong economies and their ability to build powerful armies.

Here are some of the unique racial mechanics of the Protoss:

- Warp Gate: Warp Gates are Protoss structures that allow Protoss players to warp in units directly from their Nexus. This allows Protoss players to quickly produce units at any location on the map.

- Chrono Boost: Chrono Boost is a Protoss ability that can be used to speed up the production of units or the research of technologies. Protoss players can use Chrono Boost to gain an advantage in the early game or to quickly produce a powerful army.

- Shields: Protoss units have shields, which are a layer of energy that protects them from damage. Shields regenerate over time, but they can be depleted by enemy attacks. Protoss players can use their shields to their advantage by engaging in hit-and-run tactics or by using their shields to absorb damage while their units attack.

- Psionic Storm: Psionic Storm is a Protoss spell that creates a storm of psionic energy that damages enemy units in a large area. Protoss players can use Psionic Storm to clear out enemy armies or to damage enemy structures.

Terran

The Terrans are a versatile race that combines human ingenuity with advanced technology. They have a wide variety of units to choose from, including Marines, Siege Tanks, and Battlecruisers. Terrans are also known for their powerful mobile structures, such as Command Centers and Factories.

Here are some of the unique racial mechanics of the Terrans:

- MULE: MULEs are Terran worker units that can be called down from Orbital Command Centers. M.U.L.E.s mine minerals at a faster rate than SCVs, which can help Terran players to quickly build up their economies.

- Supply Drops: Terran Supply Depots provide supply, which is needed to produce units. Terran players can also call down Supply Drops from their Command Centers, which provides additional supply. This allows Terran players to quickly build up their armies.

- Siege Tanks: Siege Tanks are Terran tank units that can transform into a siege mode. In siege mode, Siege Tanks have a longer range and deal more damage. Terran players can use Siege Tanks to defend their bases or to attack enemy bases from a distance.

- Medivacs: Medivacs are Terran flying units that can transport units and heal them. Terran players can use Medivacs to quickly move their armies around the map and to keep their units alive in battle.

Each race in StarCraft II has its own unique strengths and weaknesses. Zerg players are known for their ability to quickly produce large armies, Protoss players are known for their strong economies and powerful units, and Terran players are known for their versatility and mobile structures. The best way to learn about the unique racial mechanics of each race is to experiment with them and to play against other players.

### G.3 Typical Strategies in StarCraft II for Different Matchups

**Q1: TvZ: What's a common strategy for Terran vs. Zerg?**

**GPT3.5:** Terran vs. Zerg (TvZ):

- Bio-Mech Composition: Terran often uses a combination of infantry (Marines, Marauders) and mechanical units (Siege Tanks, Hellbats) against Zerg. This composition allows for mobility and firepower.

- Harassment: Terran players often employ drop tactics, using Medivacs to drop units into Zerg bases to disrupt their economy and force them to split their forces.

- Widow Mines: Widow Mines can be used to deter Zergling and Baneling attacks, and they can also be dropped into Zerg mineral lines to deal economic damage.

**GPT4:** A common strategy involves harassment to prevent the Zerg from comfortably expanding and massing their forces. Terrans might use Hellions to attack Zerg's worker units or dropships to move units behind enemy lines. The overall aim is to control the pace, forcing Zerg to react to Terran threats while building a strong economy and army for a decisive push.

**Claude2:**    Terran often opens with a Reaper expand into Hellions for map control and harassment. This transitions to Marine/Medivac for mobility and bio strength. Tanks added for splash against Zerglings.

**Bard:**    A common strategy for Terran vs. Zerg is to go for a macro build, such as Bio (Marines, Medivacs, and other support units) or Mech (Siege Tanks, Hellions, and other mechanical units). Terran players can also try to rush their Zerg opponent with early aggression, such as a Reaper Rush or a Marine/Marauder Rush.

### Q2: TvT: What's a typical strategy in Terran vs. Terran matchups?

**GPT3.5:**

- Tank Warfare: TvT often involves tank battles. Players set up Siege Tanks defensively and try to gain positional advantages while avoiding being caught unsieged.
- Viking Control: Air control is crucial in TvT. Vikings are used to gain air superiority and vision for Siege Tanks. Raven usage for interference matrix can also be a key factor.
- Bio vs. Mech: Terran players can choose between a bio-heavy composition (Marines, Marauders) and a mech-heavy composition (Siege Tanks, Hellions). The choice can depend on personal preference and map.

**GPT4:**    TvT often revolves around positioning and tank warfare. Siege Tanks are critical for area control, and players jockey for position and map control. Air superiority via Vikings is also crucial to provide vision and control. Since all units are available to both sides, scouting, deception, and adaptability are key.

**Claude2:**    Both players open Reaper expand for scouting information. Focus is on securing additional bases for economy. Tanks, Vikings and air control are critical for positioning and engagements.

**Bard:**    A typical strategy in Terran vs. Terran matchups is to go for a macro build, such as Bio or Mech. Terran players can also try to scout their opponent's build early and counter it with a specific composition. For example, if a Terran player sees that their opponent is going for Bio, they can counter with Siege Tanks and Hellions.

### Q3: TvP: How do Terran players generally approach a Protoss opponent?

**GPT3.5:**

- Bio Play: Terran often employs bio compositions (Marines, Marauders, Medivacs) against Protoss. Micro-intensive battles and mobility are key.
- Ghost Usage: Ghosts can snipe high-value Protoss units like Templars and Observers. They also have EMP to drain Protoss shields and energy.
- Liberator Zones: Liberators can establish zones of control and disrupt Protoss positioning with their anti-ground mode.

**GPT4:**    Terrans often use a mixture of bio units (Marines and Marauders) supplemented with Medivacs for healing and mobility against Protoss. Harassment with drops to disrupt the Protoss economy and tech is common. Terrans need to be aggressive, as a fully teched and maxed Protoss army can be difficult to defeat head-on.

**Claude2:**    Terran applies early pressure with units like Reapers, Hellions and Widow Mines. Goal is to slow Protoss economy and tech while expanding themselves. Marines and Ghosts counter late-game Protoss power units.

**Bard:**    Terran players generally approach a Protoss opponent by trying to expand their economy and build up a strong army. Terran players can also try to rush their Protoss opponent with early aggression, such as a Reaper Rush or a Hellion Rush.

**Q4: ZvZ: What's a standard strategy in Zerg vs. Zerg battles?**

**GPT3.5:**

- Zergling-Baneling Wars: ZvZ is known for Zerglings and Banelings clashing. Players aim to control these units efficiently and gain map control.
- Roach Timing Attacks: Roach-based strategies can be effective for pushing across the map and pressuring the opponent.
- Mutalisk Play: Mutalisks are used for harassment and map control in the mid-to-late game.

**GPT4:** ZvZ can be very volatile and is often characterized by early aggressive plays, like Zergling and Baneling skirmishes. Map control via scouting with Overlords is crucial to see incoming threats. Due to the mirror nature, small advantages can escalate quickly, and tech switches to counter the opponent's unit composition are common.

**Claude2:** Quickly droning up to 3 bases while producing defensive units like Queens. Tech to either Roaches for safety or Mutalisks for harassment. Late-game transition to Brood Lords.

**Bard:** A standard strategy in Zerg vs. Zerg battles is to go for a macro build, such as Mutalisk/Zergling or Roach/Hydra. Zerg players can also try to rush their opponent with early aggression, such as a Zergling Rush or a Baneling Rush.

**Q5: ZvP: What are Zerg strategies when facing Protoss?**

**GPT3.5:**

- Hydralisk and Lurker Play: Hydralisks and Lurkers are often used against Protoss, as they can deal with Gateway units effectively.
- Creep Spread: Zerg players work to spread creep to control the map and gain vision, which is crucial in ZvP.
- Tech Switches: Zerg may transition between unit compositions, such as Roach-Ravager into Brood Lords or Ultralisks, to keep Protoss players guessing.

**GPT4:** Zerg players aim to out-expand their Protoss rivals early on, using units like Zerglings and Roaches to defend against early aggression. They must be wary of Protoss area-of-effect (AoE) damage from units like High Templars and Colossi, often opting for tech like Corruptors or Vipers to neutralize key threats, and tech switches to keep Protoss guessing.

**Claude2:** Defend early Protoss pressures like Adepts while droning up. Roach/Ravager with Infestors to counter Immortals, or Hydra/Lurker for anti-air. Vipers critical for late-game.

**Bard:** Zerg strategies when facing Protoss typically involve trying to overwhelm their opponent with a large army of cheap and expendable units, such as Zerglings and Banelings. Zerg players can also try to flank their Protoss opponent with Mutalisks or to use their Hydralisks to pick off Protoss units from a distance.

**Q6: ZvT: How do Zerg players counter Terran tactics?**

**GPT3.5:**

- Baneling-Mutalisk: Zerg players often use Banelings and Mutalisks for harassment and map control. Banelings can be used to deal with Terran infantry.
- Infestors: Infestors can be used to fungal growth Terran bio units and neural parasite powerful Terran units like Siege Tanks or Thors.
- Brood Lords and Ultralisks: Late-game ZvT often involves transitioning into Brood Lords and Ultralisks for power and durability.

**GPT4:**  Zerg often faces bio-ball compositions (Marines, Marauders, Medivacs) or mechanical units (like Tanks and Thors). Zerg needs to expand quickly, defend against harassment, and may use Banelings, Zerglings, and Hydralisks for defense transitioning to more robust units like Ultralisks or Brood Lords depending on Terran composition.

**Claude2:**  Ling/Bane for Bio defense. Fast spire into Mutalisk harass. Transition to Ultralisks supported by Queens and Infestors for the often mech-heavy Terran end-game.

**Bard:**  Zerg players counter Terran tactics by using their superior mobility and economy to their advantage. Zerg players can also use their powerful units, such as Ultralisks and Brood Lords, to defeat Terran armies

## Q7: PvZ: How should Protoss approach a Zerg opponent?

**GPT3.5:**

- Gateway Expand: Protoss players often open with a Gateway expansion, using units like Adepts and Stalkers to control the early game.
- Tech Choices: Protoss can opt for Stargate play (Phoenixes or Oracles) for harassment, or they may go for a Robo facility for Immortals or Disruptors.
- High Templars: Templars are used for their Psionic Storm ability to deal with large Zerg armies.

**GPT4:**  Protoss players must secure a solid economic base while defending against Zerg's potential for rushes and harassment. Walling off chokepoints with buildings and using units like Zealots and Sentries to defend is common early game. They rely on powerful AoE units (like Colossi and High Templars) later in the game, often leveraging tech advantages for strong pushes.

**Claude2:**  Fast expand into Immortal/Archon/Chargelot to defend Zerg attacks. High Templar for storms against Hydras. Skytoss deathball critical for late game.

**Bard:**  Protoss players approach a Zerg opponent by trying to expand their economy and build up a strong army of powerful units, such as Zealots, Stalkers, and Immortals. Protoss players can also use their Colossi to deal damage to Zerg units from a distance.

## Q8: PvP: What's the strategy for Protoss vs. Protoss games?

**GPT3.5:**

- Stalker-Sentry: Early PvP often features Stalkers and Sentries for map control and defense.
- Disruptor Play: Disruptors are a key unit in PvP, used for their ability to deal massive area damage to enemy units.
- Blink Stalker Micro: Micro-management of Blink Stalkers can be a deciding factor in PvP engagements.

**GPT4:**  Early game revolves around preventing enemy harassment and scouting to anticipate tech choices. The mid-game often sees a lot of positioning battles and potential for blink Stalker harassment. Since Protoss units are powerful but costly, efficient trades and protecting key units are paramount, with late-game often seeing advanced units like Carriers or Tempests.

**Claude2:**  Robo for Immortals or fast expand with Blink Stalkers. Disruptors and High Templar for splash against massed Gateway units. Whoever gets Tempests wins the late game.

**Bard:**  The strategy for Protoss vs. Protoss games typically involves trying to scout their opponent's build early and counter it with a specific composition. For example, if a Protoss player sees that their opponent is going for a Colossi build, they can counter with Immortals and Void Rays.

**Q9: PvT: How do Protoss players generally strategize against Terran?**

**GPT3.5:**

- Adept Openings: Protoss may open with Adepts for early map control and harassment.
- Colossi and/or Storm: Colossi are strong against Terran bio, while High Templars with Storm can deal with clumped bio forces.
- Disruptors: Disruptors can be used to disrupt Terran positioning and force engagements.

**GPT4:** Protoss strategies often involve careful defense against early Terran aggression, using units like Stalkers and Sentries, while advancing tech. Harassment with units like Oracles or Phoenixes can keep Terran unbalanced. In the later game, Protoss looks to leverage high-tech units (like Colossi, High Templars, or Carriers) to break through Terran's defenses.

**Claude2:** openings to secure expand. Robo for Observer scouting and Immortals against Terran vehicles. Storms and Archons counter Marine/Marauder bio play.

**Bard:** Protoss players generally strategize against Terran by trying to expand their economy and build up a strong army of powerful units, such as Zealots, Stalkers, and Immortals. Protoss players can also use their Colossi to deal damage to Terran units from a distance.

**G.4 Ability to Outline Build Orders for Different Races**

**Q:Can you provide a standard build order for each race?**

**GPT3.5:** Terran Standard Build Order:

- Opening: Start with an SCV and send it to gather minerals.
- Supply Depot: Build a Supply Depot at 14 supply to ensure you don't get supply blocked.
- Barracks: Build a Barracks at 16 supply.
- Refinery: Take a Refinery at 17 supply for gas income.
- Orbital Command: When the Barracks is complete, upgrade your Command Center to an Orbital Command.
- Reaper: Train a Reaper from the Barracks for scouting and early map control.
- Factory: At 100 gas, build a Factory.
- Second Refinery: Take a second Refinery after the Factory.
- Starport: Build a Starport when the Factory is complete.
- Tech Lab: Attach a Tech Lab to the Barracks.
- Marine Production: Start producing Marines from the Barracks.
- Reactor: Attach a Reactor to the Factory.
- Medivac: Build a Medivac from the Starport.
- Tech Choices: Depending on your strategy, you can go for upgrades, more production facilities, or expand.

Protoss Standard Build Order:

- Opening: Start with a Probe and send it to gather minerals.
- Pylon: Build a Pylon at 14 supply to avoid getting supply blocked.
- Gateway: Create a Gateway at 16 supply.
- Assimilator: Take an Assimilator at 16 supply for gas income.
- Cybernetics Core: Build a Cybernetics Core when the Gateway is complete.
- Pylon: Construct another Pylon at around 20 supply.

- Adept or Stalker: Train an Adept or Stalker from the Gateway for early defense or map control.
- Nexus: Expand to a natural expansion at around 21 supply.
- Warp Gate Research: Research Warp Gate technology at the Cybernetics Core.
- Additional Gateways: Add more Gateways and a Robotics Facility or Stargate based on your strategy.
- Tech Choices: Depending on your strategy, go for units like Stalkers, Adepts, or Oracles, and consider building more production structures.
- Upgrades: Start researching upgrades at the Forge.
- Expand: Consider taking a third base.

Zerg Standard Build Order:

- Opening: Start with a Drone and send it to gather minerals.
- Overlord: Build an Overlord at 13 supply.
- Drones: Continue to produce Drones.
- Hatchery: At 16 supply, send a Drone to build a Hatchery at your natural expansion.
- Extractor: Take an Extractor at 17 supply.
- Spawning Pool: Build a Spawning Pool at 17 supply.
- Overlord: Create another Overlord at around 19 supply.
- Queen: When the Spawning Pool is complete, produce a Queen from your main Hatchery.
- Zerglings: Train a set of Zerglings for scouting and defense.
- Extractor: Take a second Extractor at your natural.
- Evolution Chamber: Build an Evolution Chamber to start upgrades.
- Additional Queens: Create more Queens for larva injects and creep spread.
- Tech Choices: Depending on your strategy and opponent, go for Roaches, Banelings, or additional bases.
- Lair: Start a Lair tech structure when you have the resources.
- Further Expansion: Consider expanding to a third base.

These build orders provide a foundation for each race, but remember that flexibility and adaptation are key in StarCraft II. Adjust your build order based on scouting information and your opponent's actions to maximize your chances of success.

**GPT4:** Certainly, "StarCraft II" revolves heavily around build orders, which are pre-determined sets of actions aimed at maximizing efficiency, economy, and production. These build orders set the stage for your mid-game and late-game strategies. Below are standard build orders for each race, which are foundational and can be adapted based on opponents' actions. Remember, the numbers represent the total worker count you should have when you begin each step.

Terran (Standard Opening):

- 14/15: Supply Depot
- 16/23: Barracks
- 16/23: Refinery
- 19/23: Orbital Command (immediately after Barracks completion)
- 20/23: Reaper (for scouting and slight harassment)
- 21/23: Command Center (on the low ground if you're feeling safe)
- 21/23: Factory (after Reaper starts)
- 23/31: Supply Depot

- Follow-up: Depends on scouting information (could include more Barracks, a Starport, or additional expansions and tech buildings).

Zerg (Hatch First Opening):

- 13/14: Overlord
- 17/22: Hatchery (expand)
- 18/22: Extractor (gas)
- 17/22: Spawning Pool
- 19/22: Overlord
- 20/22: Metabolic Boost (Zergling speed, once you have 100 gas)
- 21/30: Queen x2 (once Spawning Pool is done)
- 30/36: Zerglings or other defensive units depending on what you scout
- Follow-up: Based on the opponent's build, you might add additional bases, tech up (e.g., Roach Warren, Lair), or build more combat units.

Protoss (Standard Gateway Expand):

- 14/15: Pylon
- 16/23: Gateway
- 17/23: Assimilator
- 20/23: Nexus (expand)
- 20/23: Cybernetics Core
- 21/31: Assimilator
- 22/31: Pylon
- 23/31: Adept or Stalker (after Cybernetics Core finishes for defense/scouting)
- Follow-up: Varied based on scouting but might include additional Gateways, a Robotics Facility, or a Stargate, and tech advancements like Warp Gate research.

These build orders are basic openings designed for beginners to intermediates and are meant to establish your economy while keeping you safe in the early game. They are not rigid play patterns. "StarCraft II" is a dynamic game, and all strategies need to adapt to opponent actions. Effective scouting and understanding of your opponent's strategy are crucial, and as you gain more experience, you'll begin to learn how to alter your build order in response to what you're seeing on the battlefield.

**Claude2:** Here are some standard opening build orders for each race in StarCraft II:

Terran Build Order:

- 14 Supply Depot
- 16 Barracks
- 16 Refinery
- @100% Barracks - Reaper and Orbital Command
- 22 2nd Command Center
- @400 minerals - 2nd Barracks

This gets an early Reaper for scouting and harassment, while using the Orbital Command for extra economy and flexibility. The 2nd CC and Barracks enable a solid 2-base infrastructure.

Zerg Build Order:

- 17 Spawning Pool
- 17 Extractor

- 17 Hatchery
- @100% Spawning Pool - 2 Queens
- 21 3rd Hatchery
- @100 gas - Metabolic Boost (Zergling speed)
- @100% 3rd Hatch - 2 more Queens

This provides a quick Queen defense, fast Zergling speed, and lots of Larva for droning with the fast 3 Hatcheries.

Protoss Build Order:

- 14 Pylon
- 16 Gateway
- 17 Assimilator
- 20 Nexus
- 20 Cybernetics Core
- 22 Pylon
- @100% Cybernetics Core - Adept (or Stalker)

This gets an expand into early tech for defense and scouting. The Adept provides psionic pressure and intel gathering. The build has potential to transition into several tech paths.

Let me know if you need any clarification or have additional questions! Providing build orders is a great way to demonstrate StarCraft knowledge.

**Bard:**   Sure, here is a standard build order for each race in StarCraft II: Terran

Bio Build Order:

- 17 Supply Depot
- 18 Barracks
- 19 Refinery
- 20 Supply Depot
- 21 Barracks
- 22 Refinery
- 23 Factory
- 24 Supply Depot
- 25 Command Center (Second base)
- 26 Refinery
- 27 Engineering Bay
- 28 Supply Depot
- 29 Refinery
- 30 Supply Depot
- 31 Starport
- 32 Supply Depot

Mech Build Order:

- 17 Supply Depot
- 18 Refinery
- 19 Supply Depot
- 20 Factory

- 21 Supply Depot
- 22 Refinery
- 23 Barracks
- 24 Refinery
- 25 Command Center (Second base)
- 26 Refinery
- 27 Supply Depot
- 28 Engineering Bay
- 29 Supply Depot
- 30 Refinery
- 31 Supply Depot
- 32 Starport

Zerg

Mutalisk/Zergling Build Order:

- 17 Hatchery
- 18 Gas
- 19 Spawning Pool
- 20 Extractor
- 21 Overlord
- 22 Extractor
- 23 Overlord
- 24 Lair
- 25 Overlord
- 26 Overlord
- 27 Hydralisk Den
- 28 Overlord
- 29 Overlord
- 30 Overlord
- 31 Spire

Roach/Hydra Build Order:

- 17 Hatchery
- 18 Gas
- 19 Spawning Pool
- 20 Extractor
- 21 Overlord
- 22 Extractor
- 23 Overlord
- 24 Lair
- 25 Overlord
- 26 Overlord
- 27 Hydralisk Den
- 28 Overlord

- 29 Evolution Chamber

- 30 Overlord

- 31 Overlord

Protoss

Zealot/Stalker Build Order:

- 17 Gateway

- 18 Probe

- 19 Pylon

- 20 Forge

- 21 Gateway

- 22 Cybernetics Core

- 23 Pylon

- 24 Gateway

- 25 Gateway

- 26 Gateway

- 27 Stargate

- 28 Pylon

- 29 Pylon

Colossus Build Order:

- 17 Gateway

- 18 Probe

- 19 Pylon

- 20 Forge

- 21 Gateway

- 22 Cybernetics Core

- 23 Pylon

- 24 Gateway

- 25 Gateway

- 26 Gateway

- 27 Robotics Bay

- 28 Pylon

- 29 Pylon

These are just standard build orders, and you can adjust them to fit your own playstyle. It is also important to scout your opponent's build early and to counter it with a specific composition.

### G.5 Classic Strategies and Counterplays in StarCraft II

**Q1: PvZ: What are effective strategies and counterplays against a roach-hydra attack?**

**GPT3.5:**  PvZ: Against Roach-Hydra Attack:

Strategy for Protoss:

- Force Fields: Use Sentries to create Force Fields to segment the Roach-Hydra army, preventing it from engaging as a whole.
- Colossi: Colossus units deal area damage and are effective against clumps of Roaches and Hydralisks.
- Immortals: Immortals are strong against Roaches due to their bonus damage against armored units.
- Archons: Archons have splash damage and can help clear out Hydralisks.

Counterplay for Zerg:

- Tech to Hive: Transition to Hive tech for units like Vipers, which can abduct key enemy units or use Blinding Cloud to reduce the effectiveness of enemy ranged units.
- Hydra Speed Upgrade: Research Muscular Augments for Hydralisks to improve their mobility.
- Burrow: Use Burrow for Roaches to save them from being picked off and to regain health.
- Drop Play: Drop Roaches or Hydralisks into the Protoss main base for harassment to divert their attention. Creep Spread: Maintain good Creep spread to ensure mobility and vision.

**GPT4:**  When Protoss players face a Zerg opponent focusing on Roach-Hydra, they're dealing with a versatile force that can be potent in a head-on fight due to its range, damage, and health, particularly in the mid-game.

- Strategic Positioning: Force engagements in narrow passages where the Zerg cannot leverage their numbers. Simultaneously, avoid open areas where the Zerg army can fully surround your forces.
- High Templars: These units are crucial. Their Psionic Storm ability deals massive damage to biological units, which makes them particularly effective against Hydralisks and clumped-up Roaches.
- Colossi: With their extended thermal lance upgrade, they can exploit their superior range and area-of-effect damage to decimate tightly packed Roach-Hydra armies.
- Sentries: Force Fields can cut off portions of the Zerg army, preventing them from engaging all at once, or Guardian Shield can reduce incoming ranged damage.
- Harassment: Divide the Zerg's attention. Warp Prisms, Zealot run-bys, and Dark Templars can force the Zerg player to split their army, making it less effective in a direct engagement.

**Claude2:**  Against Roach-Hydra, go Immortal/Archon with chargelots.  Immortals counter Roaches, Archons counter Hydras. Focus fire with Immortals.alternatively, fast tech to Colossus. Use Sentries for forcefields to split Zerg army. Expand behind and get High Templar for storms.

**Bard:**  Effective Strategies:

- Go for a Colossus build. Colossi are very effective against Roaches and Hydras, and they can also deal damage to Zerg structures.
- Use Void Rays to pick off Roaches and Hydras from a distance. Void Rays are also very effective against Zerg Overlords, which can help you to deny your opponent vision and to make it more difficult for them to attack you.
- Use Warp Gates to quickly produce units and to respond to Zerg attacks. Warp Gates allow you to warp in units directly from your Nexus, which can help you to quickly build up your army and to defend your base. Effective Counterplays:
- Use Observers to scout your opponent's base and to see what kind of army they are building. This will give you time to prepare for their attack and to build the appropriate counter-units.

- Use Pylons to power your units and to build Shielding Batteries. This will help to protect your units from Zerg attacks.
- Use Force Fields to block off Zerg units and to create chokepoints. This will make it more difficult for your opponent to attack your base.

**Q2: TvZ: How should Terran approach a mutalisk-zergling-baneling composition? Are there gambits or counter-strategies to consider?**

**GPT3.5:** TvZ: Against Mutalisk-Zergling-Baneling Composition:

Strategy for Terran:

- Thors: Thors are strong anti-air units that can target Mutalisks effectively.
- Widow Mines: Widow Mines can deal significant damage to clumps of Mutalisks if properly positioned.
- Marine Splitting: Split your Marines to minimize the impact of Baneling detonations.
- Missile Turrets: Build Missile Turrets in key locations to protect against Mutalisk harassment.
- Vikings and Liberators: Vikings can help control the air, and Liberators can provide anti-ground support.

Counterplay for Zerg:

- Baneling Drops: Use Overlords to drop Banelings onto clumps of Marines.
- Transition to Ultralisks: Ultralisks are powerful melee units that can be effective against Terran infantry.
- Corruptors: Corruptors can be used to target Thors and Vikings and reduce the anti-air capabilities of the Terran.
- Infestors: Fungal Growth can be used to lock down Marines and other units.
- Creep Spread: Maintain Creep spread to gain map control and mobility.

**GPT4:** The Muta-Ling-Bane composition is a highly mobile harassment force that seeks to control the map, prevent expansions, and pick off stray units. Dealing with it requires a mix of solid defense and map awareness.

- Marine-Medivac-Widow Mine: Marines, with their high DPS, are effective against Zerglings and Mutalisks, especially when combined with Medivacs for healing. Widow Mines add a layer of defense against Banelings and can devastate clumped Mutalisks.
- Thors and Liberators: Thors can help deter Mutalisks with their strong anti-air attack, especially in High Impact Payload mode. Liberators can zone out Mutalisks and provide support against ground units.
- Base Defense: Well-placed Missile Turrets and Bunkers can help defend against harassment. Building walls at choke points prevent Zergling run-bys.
- Map Control: Establish map control with Marine or Hellion squads and maintain good vision to see enemy movements. This approach helps in preparing for incoming attacks and catching the Zerg army out of position.

**Claude2:** Marines, Thors and Widow Mines counter Mutalisks. Thors splash damage and Widow Mine hits punish clumped Mutalisks. Turrets at mineral lines prevent economic harassment. Bio-mine with Medivacs can pick apart Zerglings. Siege Tanks zone out Banelings.

**Bard:** Effective Strategies:

- Go for a Bio build. Bio units are very effective against Mutalisks and Zerglings, and they can also deal damage to Zerg structures.

- Use Siege Tanks to deal damage to Mutalisks and Zerglings from a distance. Siege Tanks are also very effective against Zerg Overlords, which can help you to deny your opponent vision and to make it more difficult for them to attack you.

- Use Medivacs to heal your units and to transport them around the map. Medivacs can also be used to drop your units into the middle of a Zerg army, which can be a very effective way to engage your opponent. Effective Counterplays:

- Use Missile Turrets and Bunkers to defend your base from Mutalisk attacks. Missile Turrets and Bunkers are both very effective against Mutalisks, and they can also be used to defend your workers and your structures.

- Use Ravens to detect Mutalisks and to cast Interference Matrix on them. This will make it more difficult for your opponent to control their Mutalisks and to attack you.

- Use Vikings to shoot down Mutalisks. Vikings are the only Terran unit that can attack air units without moving, which makes them very effective against Mutalisks.

**Q3: TvP: When faced with an archon-zealot push, what are the best strategies and counters for Terran?**

**GPT3.5:** TvP: Against Archon-Zealot Push:

Strategy for Terran:

- Kite and Focus Fire: Kite the Zealots with your ranged units (e.g., Marines and Marauders) and focus fire on Archons.
- Medivac Micro: Use Medivacs to pick up and reposition units to avoid the melee engagements.
- Widow Mines: Position Widow Mines to hit the Zealots and soften the Archons.
- Ghosts: EMP from Ghosts can drain the shields of Protoss units, including Archons.

Counterplay for Protoss:

- Force Fields: Create Force Fields to segment the Terran army and prevent kiting.
- Immortals: Immortals are strong against Archons due to their bonus damage against armored units.
- High Templars: Use Psi Storm to deal AoE damage to the Terran infantry.
- Air Units: Consider incorporating air units like Void Rays or Phoenixes for added versatility.
- Guardian Shield: Use the Guardian Shield ability from Sentries to reduce incoming ranged damage.

Remember that these strategies and counterplays are not exhaustive, and adaptability is key in StarCraft II. The specific outcome of battles can depend on micro-management, unit composition, upgrades, and positioning, so always be prepared to adjust your tactics based on the evolving situation in the game.

**GPT4:** Archon-Zealot (often with some Immortals) is a powerful, close-quarters Protoss army composition that relies on brute force and strong front-line units.

- Ghost Usage: Ghosts' EMP Round is highly effective in draining the shields of Archons, Zealots, and Immortals, significantly weakening their upfront staying power in battles.
- Liberators and Siege Tanks: The use of Liberators with Defender Mode and Siege Tanks can create a zone of control that Zealots and Archons cannot approach easily. Kiting into these zones can decimate a Protoss army that's chasing.
- Micro-intensive Kiting: Using bio forces (Marines and Marauders) supplemented with Medivacs for healing, a player can continuously kite against Zealots and Archons, which lack ranged attacks.

- Sim City: Terrans can build in a way that chokes off paths (using Supply Depots, Bunkers, etc.), restricting the mobility and attack angles of melee units like Zealots and forcing Archons to funnel through tight spaces where they can't leverage their splash damage.

Each of these scenarios emphasizes not just the composition of your forces but also the importance of control, positioning, adaptability, and game sense. "StarCraft II" is a game of constant decision-making, and successful players must be able to read their opponents and adjust their strategies on the fly.

**Claude2:** Marines and Marauders counter Zealots, while Ghosts and Liberators counter Archons. Kite with bio and emp Archons. Siege Tanks in tank mode punish Zealot chargelots. Well-spread mines damage clumped Zealots. Scan aggressively to preemptively snipe Observers and enable Widow Mines.

**Bard:** Effective Strategies:

- Go for a Bio build. Bio units are very effective against Zealots and Archons, and they can also deal damage to Protoss structures.
- Use Siege Tanks to deal damage to Zealots and Archons from a distance. Siege Tanks are also very effective against Protoss Colossus, which can help you to defend your base and to make it more difficult for your opponent to attack you.
- Use Medivacs to heal your units and to transport them around the map. Medivacs can also be used to drop your units into the middle of a Protoss army, which can be a very effective way to engage your opponent.

Effective Counterplays:

- Use Missile Turrets and Bunkers to defend your base from Archon and Zealot attacks. Missile Turrets and Bunkers are both very effective against Zealots and Archons, and they can also be used to defend your workers and your structures.
- Use Scans to detect Protoss units and to see what kind of army they are building. This will give you time to prepare for their attack and to build the appropriate counter-units.
- Use Snipers to pick off Protoss units from a distance. Snipers are very effective against Zealots and Archons, and they can also be used to deal damage to Protoss Colossus.

# H NeurIPS Paper Checklist

## H.1 Claims

Answer: [Yes]

Justification: The abstract and introduction clearly outline the main claims and contributions of developing a new method for integrating LLMs into RTS gaming, particularly StarCraft II, and improving model interpretability. These sections accurately reflect the scope of the research and its applicability to other complex decision-making environments

## H.2 Limitations

Answer: [Yes]

Justification: TextStarCraft II's limitations include its reliance on non-visual data inputs, the use of scripted micro policies, and constrained performance due to resource limitations. For more detailed information, please refer to the Discussion section 8.

## H.3 Theory Assumptions and Proofs

Answer: [NA]

Justification: The paper primarily focuses on experimental results and method development rather than theoretical proofs.

## H.4 Experimental Result Reproducibility

Answer: [Yes]

Justification: The paper details all necessary configurations, methodologies, and the TextStarCraft II environment used, ensuring reproducibility. Additionally, we have open-sourced the code, data, and a demo, providing full access for researchers to replicate and verify our results comprehensively.

## H.5 Open access to data and code

Answer: [Yes]

Justification:We have provided open access to our complete source code and dataset, ensuring full reproducibility of our experiments. All resources, including training and testing scripts, are available in a publicly accessible repository linked in the supplemental materials of our paper. We also provide the anonymous link:`https://anonymous.4open.science/r/Large-Language-Models-play-StarCraftII-8C45/readme.md`.

## H.6 Experimental Setting/Details

Answer: [Yes]

Justification: Our experimental setup is thoroughly detailed in Section A.1, including comprehensive descriptions of agent and opponent selection, map choice, parameter settings, and more, ensuring a robust and reproducible experimental environment.

## H.7 Experiment Statistical Significance

Answer: [Yes]

Justification: Our paper presents error bars calculated as the standard deviation across multiple experiment runs, capturing variability due to initialization. These error bars are clearly displayed and annotated in the experimental result tables and figures.

## H.8 Experiments Compute Resources

Answer: [Yes]

Justification: we have introduced our experiments compute resources, detailed in A.3.

## H.9 Code Of Ethics

Answer: [Yes]

Justification: Our research complies with the NeurIPS Code of Ethics, ensuring ethical standards and transparency in all experimental procedures, with measures in place to mitigate any potential risks, detailed in 9.

## H.10 Broader Impacts

Answer: [Yes]

Justification:The paper discusses both positive and negative societal impacts of using LLMs in strategic environments, detailed in 9.

## H.11 Safeguards

Answer: [NA]

Justification: Our research does not introduce data or models with a high risk for misuse. The project focuses on developing strategic AI capabilities within the controlled environment of StarCraft II using TextStarCraft II, which poses no inherent risk of dual-use or misuse. Thus, no specific safeguards beyond standard ethical research practices were required.

## H.12 Licenses for existing assets

Answer: [Yes]

Justification: All used assets are properly credited with detailed references and URLs. We explicitly state each asset's license type, predominantly CC-BY 4.0, ensuring all terms of use are respected and documented as per compliance standards.

## H.13 New Assets

Answer: [Yes]

Justification: We introduced TextStarCraft II, a new, thoroughly documented asset. Documentation including setup, usage, and licensing details is available in a publicly accessible, anonymized repository, ensuring transparency and integrity in the review process. `https://anonymous.4open.science/r/Large-Language-Models-play-StarCraftII-8C45/readme.md`

## H.14 Crowdsourcing and Research with Human Subjects

Answer: [Yes]

Justification: We involved 30 StarCraft II masters and grandmasters, including professional and semi-professional players, in our study. For more details on the participant engagement and data handling, please see the section referenced in A.4.

## H.15 Institutional Review Board (IRB) Approvals or Equivalent for Research with Human Subjects

Answer: [NO]

Justification:Although our study involved human participants, their engagement was restricted to game-related QA and gameplay testing, which are low-risk activities typical of gaming studies. This type of interaction generally does not require IRB approval as it does not involve exposure to physical or psychological risks beyond those encountered in normal gameplay. Participants were fully briefed on the nature of the tests and their voluntary participation was secured, aligning with ethical standards for such research.

