# OpenReview forum: "Large Language Models Play StarCraft II:Benchmarks and A Chain of Summarization Approach"
_NeurIPS.cc/2024/Conference — NeurIPS 2024 poster_

### Official Review · Reviewer_gmAe · 2024-07-04

**Soundness:** 2
**Presentation:** 2
**Contribution:** 2
**Rating:** 5
**Confidence:** 4

**Summary:**

This paper introduces TextStarCraft II, a text-based environment to evaluate the strategic decision-making and planning capabilities of large language models (LLMs) in real-time scenarios within StarCraft II (SC2). The study addresses the limitations of traditional Chain of Thought (CoT) methods by proposing the Chain of Summarization (CoS) method, which enhances LLMs’ abilities to process complex information and make strategic decisions efficiently. Key experiments include testing various LLMs against SC2's built-in AI, evaluating commercial models on SC2 knowledge, and conducting human-AI matches to assess performance and strategic adaptability.

**Strengths:**

- Chain of Summarization (CoS) Method: The introduction of the CoS method improves LLMs' ability to summarize and process complex game states, leading to more effective decision-making in real-time scenarios.

- Impact on AI Research: The study’s findings contribute to the broader field of AI research by providing insights into the capabilities and limitations of LLMs in handling complex, real-time decision-making tasks.

**Weaknesses:**

- Human Interaction Assurance: The paper does not clearly explain how LLMs interact with the game. From lines 120-121, it seems that human players might use LLM suggestions to operate the game. If this is the case, how do you ensure that players fully adhere to the LLM's suggestions?
- Latency and Real-Time Feedback: In Section 5.3, detailed information on latency is not provided. Can your method provide real-time feedback?
- Resource Dependency: The reliance on rule-based scripts for micro-management and the limitation to text-based inputs may restrict the diversity and applicability of AI strategies.

**Questions:**

N/A

---

> ### Author Rebuttal · Authors · 2024-08-07
>
> Dear Reviewer,
>
>
> Thank you for your thorough review and insightful comments. Below, we address your specific concerns and outline the changes we plan to make in the revised version.
>
> **Q1**:Human Interaction Assurance
>
> We apologize for any confusion caused by our unclear explanation. To clarify, in our TextStarCraft2 environment, the LLM agent interacts directly with the game environment without human intervention. The LLM receives text-based observations of the game state and outputs text-based decisions, which are then automatically translated into game actions by our system. There is no human intermediary executing LLM decisions. In contrast, LLM follows its own suggestions to decides which action to do (such as "Build Cybernetics Core","Attack enemy base").
>
> We will revise this section in our paper to provide a more detailed explanation of how LLMs interact with the game environment. Thank you for pointing out the clarity issue.
>
> **Q2**:Latency and Real-Time Feedback
>
> The latency in our system is primarily determined by the inference speed and model size of the LLM. By leveraging fine-tuned open-source large language models (LLMs), our approach can achieve real-time feedback. Our Chain of Summarization (CoS) method is crucial in enabling this real-time feedback, significantly outperforming traditional Chain of Thought (CoT) methods in terms of responsiveness.
>
> For instance, with CoS, even smaller LLMs like the fine-tuned Qwen-2 7B can achieve real-time interaction (100 ms/step) and demonstrate competitive performance against human players. This level of real-time capability is not possible with standard CoT methods, highlighting the effectiveness of our CoS approach in facilitating swift strategic decision-making in complex, dynamic environments.
>
> Although larger models such as GPT-4, Claude, and Gemini still face challenges in achieving real-time feedback (1s/step) due to their size, complexity, and network transmission factors, our CoS method significantly reduces their response times compared to CoT. This improvement allows for near-real-time performance in many scenarios. We are continuously optimizing our system to further enhance the real-time capabilities of these larger models.
>
> Below are the delays observed for each step when different LLMs interact with the CoS method:
>  | Model Type     | Model Name   | Delay (each step) |
> |----------------|--------------|-------------------|
> | Finetuned Open-Source LLM | Qwen2-7b      | 98ms             |
> | Finetuned Open-Source LLM | Qwen2-1.8b    | 64ms             |
> | Finetuned Open-Source LLM | LLAMA2-7B    | 102ms            |
> | Closed-LLM     | GPT3.5       | 1.2s             |
> | Closed-LLM     | GPT4         | 2.3s             |
> | Closed-LLM     | Gemini-PRO   | 0.3s             |
>
> **Q3**:Resource Dependency
>
> Our LLM utilizes macro actions such as training units, constructing buildings, and researching technologies—commonly known in the StarCraft II community as "build orders." While rule-based scripts handle some micro-level actions, they do not constrain strategic or tactical variety.
>
> In our experiments, LLM agents have successfully executed diverse strategies. For example:
>
> - Mass Void Rays: A strategy focusing on producing a large number of powerful air units.
> - Stalker-Colossus: A balanced army composition combining ground and support units.
> - Carriers: A high-tech strategy centered around powerful capital ships.
>
> These examples demonstrate the LLM's ability to adapt unit compositions and respond to potential threats dynamically, showcasing a wide range of strategic approaches without limiting the AI's ability to innovate.
>
> Regarding the limitation to text-based inputs, we consider that focusing solely on text allows us to better assess the core capabilities of language models without the need to consider their abilities in processing other modalities like vision or speech. By expressing the strategic decision-making process through text interactions, we can concentrate more on evaluating the language models' strategic reasoning and planning abilities in complex and dynamic environments.
>
> On the other hand, we also recognize the potential benefits of incorporating visual information. We are actively exploring the integration of vision-language models (VLMs) into our framework. This multimodal approach could enhance the model's overall ability to process information and make AI decisions, potentially leading to more sophisticated and adaptable strategies.

---

> > ### Comment · Reviewer_gmAe · 2024-08-08
> >
> > Thank you for the authors' response.
> > We have reviewed your feedback and noted that most of our concerns have been addressed.

---

### Official Review · Reviewer_Q8qA · 2024-07-14

**Soundness:** 3
**Presentation:** 3
**Contribution:** 4
**Rating:** 8
**Confidence:** 3

**Summary:**

The authors introduce TextStarCraft II, a framework for transferring the state of the Real Time Strategy (RTS) Starcraft II (SC2) into text form and Chain of Summarization (CoS), an extension of the traditional Chain of Thought (CoT) style prompting for condensing information and accelerating LLM inference. The authors demonstrate their system is capable of defeating gold-level, human players in SC2 in real time matches.

**Strengths:**

The authors introduce a novel benchmark environment for LLMs in the form of TextStarCraft II, an environment that allows for a rigorous evaluation of an LLM’s ability to do real-time, complex planning. They further introduce a new prompting method, CoS, to allow for information compression and multi-step inference to combat the otherwise prohibitively slow inference speed of LLMs when it comes to a RTS game. The performance of their approach when playing against the pre-programmed AIs and human-players proves the effectiveness of their approach.

**Weaknesses:**

A weakness of the authors’ approach is that there is a high likelihood that most modern LLMs were trained on text that included comprehensive strategy discussions of SC2 in text form. However, this is mostly mitigated by the authors’ investigation of both open and closed-source LLMs in section 5.1. However, it would still be interesting to see if these results still hold in newer RTS games that are likely to not be found in most LLMs’ training distribution.

A lesser weakness the reviewer would like to suggest to the authors is that it may be worth considering staying away from Starcraft-specific terms in order to reach a wider audience that may be less familiar with the Starcraft series as a whole.

**Questions:**

1.) When playing against human players, where any restrictions placed on availability of units or maps?

2.) Do the authors believe the results would generalize across the RTS genre and has there been any investigation into this?

3.) To what extent were the macro and micro actions predefined?

**Limitations:**

The authors briefly touch on some limitations of their work, however it is unclear whether the topics addressed in the questions were not included due to being outside the scope of the work, or not relevant given some metric or detail the reviewer may have missed.

---

> ### Author Rebuttal · Authors · 2024-08-07
>
> Thank you for your valuable feedback. Below is our response to your inquiry.
>
>  **Weaknesses**
>
> We appreciate your suggestion about Starcraft-specific terminology. In our revised version, we will strive to make our paper more accessible to a broader audience by clarifying or simplifying Starcraft-specific terms where possible.
>
>
> **Q1**:  Restrictions in Human vs. LLM Agent Matches
>
> In our experiments, we maintained a full-range StarCraft II environment with no restrictions on maps, units, or game versions. We utilized 14 different maps from both the 2022 and 2023 ladder seasons and tested across two game versions (5.0.11 and 5.0.12). Human players engaged with LLM agents using standard input devices (keyboard and mouse) under typical 1v1 game settings, similar to the well-known AlphaStar vs. MaNa show matches.
>
> At the same time, to ensure fairness in the game, we make sure that the LLM agent has access to the same game information as humans, for example:
>
> 1. Map Visibility: The LLM agent operates under the same fog of war constraints as human players. It does not have access to any information that would be hidden from a human player in a standard game, ensuring equal information availability for both sides.
> 2. Decision Frequency: We calibrated the LLM agent's decision-making frequency to be comparable to that of human players. This means the AI is not making decisions at a superhuman rate, but rather at intervals similar to what a skilled human player might achieve.
>
> These limits ensure our AI competes fairly with humans. The focus is on testing strategic thinking and decision-making, not mechanical advantages.
>
>
> **Q2**: Generalization Across the RTS Genre
>
>
> We agree that this is an interesting and valuable observation. Testing LLM capabilities on newer RTS games not included in their training data would indeed provide a fairer assessment and better reflect the models' adaptability to out-of-distribution tasks. While creating a benchmark framework for newer RTS games is more challenging, we are actively working on extending our framework to more complex, recent RTS environments(Such as Storm Gate and Battle ACE). We look forward to sharing these results in future work and welcome your continued interest in this direction.
>
> **Q3**: Predefined Macro and Micro Actions
>
>
> 1. We predefine macro actions as follows:
> - Building Construction: The LLM determines which buildings to construct based on its strategic analysis and current game state. This decision-making process follows established community conventions known as "build orders," which can be found at https://lotv.spawningtool.com/.
> - Unit Production: The LLM decides which units to produce, demonstrating its understanding of army composition and resource management.
> - Research and Upgrades: The LLM makes decisions on which technologies to research and upgrades to pursue, aligning with its overall strategy.
> - High-level Military Strategy: The LLM determines when to scout, defend, or launch attacks based on its assessment of the game situation.
> 2. Micro Actions and Execution: While the LLM focuses on high-level decision-making, rule-based scripts handle the execution details, for example:
> - Building Placement: Once the LLM decides to construct a building, rule-based scripts interact with the environment to execute the construction.
> - Unit Training Facilities: After the LLM decides which units to produce, scripts select the appropriate facilities to carry out these orders.
>
> By separating strategic decisions from tactical execution, our framework mirrors real-world scenarios where high-level planning is distinct from operational implementation.
>
> **Limitation**
>
> Regarding the limitations of our work, we will expand and clarify the following points in our revised version:
>
> 1. Generality: We will discuss how our methods can be applied to other RTS games and mention that we are extending our framework to more complex games such as "Storm Gate" and "Battle ACE" to test the adaptability of LLMs.
>
> 2. Decision and Execution Balance: We will further explain the balance between LLM decision-making and rule-based execution, describing the separation of macro decisions (e.g., building selection, unit production) from micro execution (e.g., building placement, unit training) and their effectiveness in complex strategic environments.
>
> 3. Resource Limitations: We will describe the impact of resource limitations on system performance and introduce our plans to optimize resource usage, including exploring more efficient computing resources to improve overall performance and real-time decision-making capabilities.
>
> Thank you again for your insightful comments and feedback, which are crucial for improving the quality of our research. We look forward to continuing this dialogue and further enhancing the contributions of our work.

---

> > ### Comment · Reviewer_Q8qA · 2024-08-13
> >
> > Thank you for your explanations. I'll keep my current score.

---

### Official Review · Reviewer_B3oZ · 2024-07-17

**Soundness:** 3
**Presentation:** 3
**Contribution:** 3
**Rating:** 6
**Confidence:** 4

**Summary:**

This paper introduces TextStarCraft II, a benchmark designed to assess long-term strategic decision-making in the context of playing StarCraft II. It models the game-playing process through pure textual representation and also develops a chain-of-summarization pipeline to aid LLM's decision-making to achieve victory. They evaluate various LLMs under different prompting strategies against the built-in AI mode of StarCraft II. The experimental results indicate that prompting powerful closed-source LLMs (e.g., GPT-4) and fine-tuned open-source LLMs (e.g., QWEN-7B) can perform competitively with the Level 5 (the highest level) built-in AI of StarCraft II.

**Strengths:**

- The topic is interesting.
- The contribution of proposing a long-term strategic decision making benchmark for evaluating LLMs is beneficial to the community.
- The evaluation is extensive

**Weaknesses:**

- The objective of how LLMs achieve victory in the game is not clearly defined.

- The process by which LLMs execute actions and make observations within the textual setting is insufficiently explained.

**Questions:**

The paper introduces a benchmark for evaluating large language models' (LLMs) long-term strategic decision-making capabilities in the context of playing StarCraft II. This benchmark models gameplay in a purely textual environment and includes the design of a chain-of-summarization pipeline to aid LLMs in playing and winning the game. The topic is interesting and beneficial to the LLM research community. However, I have several concerns, primarily about the overall pipeline setting and the mechanics of how LLMs engage with the game.
- The objective of optimizing LLMs through prompting or fine-tuning to win the game appears to be inadequately defined. It remains unclear what the specific winning criteria for LLMs are and how these models compete against the built-in AI to meet these criteria.
- The methodology detailing how LLMs execute actions and observe the environment in each round lacks clarity. According to Figure 1, each round involves the LLM making decisions regarding necessary actions and translating multi-frame observations into text. However, it is not explicitly described what types of actions are executed or how multi-frame observations are represented within the purely textual environment.
- The experimental results indicate that well-optimized LLMs, with appropriate prompting, can perform competitively with built-in AIs. It would be beneficial to provide deeper insights.

---

> ### Author Rebuttal · Authors · 2024-08-05
>
> Apologies for the lack of clarity in our methodology description.
>
> **Q1**: Standard for Achieving Victory
>
> In our TextStarCraft2 environment, the victory conditions mirror those of a standard StarCraft II 1v1 game, consistent with established AI benchmarks such as AlphaStar, ROA-Star, or DI-Star. To secure a victory, the LLM agent must destroy all of the built-in AI structures. This criterion is a standard setting across various StarCraft II AI competitions, ensuring that our benchmark aligns with common practices in the field.
>
> To beat the built-in AI, the LLM must engage in a complex series of long-term strategic decisions. This includes resource gathering, constructing buildings, developing technologies, and producing combat units. The LLM needs to outperform the built-in AI over an average game duration of approximately 20 minutes (7000 steps). This process typically involves intricate resource allocation, tactical maneuvering, and unit countering strategies, among other sophisticated decisions.
> In our study, we observed some interesting behaviors from LLM agent playing a strategy game. LLM agent was controlling one faction (called Protoss) against a built-in computer opponent controlling another faction (called Zerg). Here are two notable scenarios:
> 1. Predicting and Preparing for Attacks(Figure 3.b):
>   - The LLM agent scouted the enemy base and noticed a large army.
>   - It predicted an incoming attack and decided to build defensive structures.
>   - About a minute later, when the enemy attacked, the LLM agent was well-prepared with both defensive buildings and its own army.
> 2. Adapting Army Composition(Figure 7):
>   - In the first encounter, the LLM agent's initial army (mostly ground units, zealots) struggled against the enemy's forces(roaches and hydras).
>   - LLM agent then decided to change its strategy, producing more flying units to counter the enemy's ground-based army.
>   - This change in tactics allowed LLM agent to barely survive the first attack wave.
>   - By the time the enemy launched a second attack, LLM agent had built up enough flying units to defend its territory effectively.
>
> **Q2**: Action and Observation Mechanisms
>
> **Multi-frame Summarization**: After each K steps, we collect several frames of game state information, each represented as a textual description. In each round of the game:
>
>   1. We represent each frame as the following text information:
>
>     - Resources (e.g., "Minerals: 500, Vespene Gas: 200")
>     - Buildings (e.g., "2 Gateway, 1 Nexus")
>     - Units (e.g., "15 Zealots, 5 Carriers")
>     - In-process activities (e.g., "Researching Warpgate: 70% complete")
>     - Visible enemy units and structures
>     - Current research status
>
>   Then, single-frame summarization step exacts important information(like supply, current army composition) for each frame.
>   After that, multi-frame approach helps capture temporal dynamics and ongoing processes. We then prompt the LLM to summarize these multiple textual representations into a concise, structured format. This process is designed to mimic how human players process and prioritize game information.
>
>   2. Summarization Process: The LLM is instructed to summarize the multi-frame information in the following structured manner:
>
>     1. Game Overview: A snapshot of key metrics (e.g., game time, worker count, resources, supply).
>     2. Current Game Stage: An assessment of the game's progression.
>     3. Our Situation:
>       - Units and Buildings: A summary of our army composition and structures.
>       - Economy: An evaluation of our economic status.
>       - Technology: An overview of our technological advancements.
>     4. Our Strategy: An interpretation of our current strategic position.
>     5. Enemy's Strategy: An analysis of the opponent's apparent strategy based on observable information.
>     6. Key Information: Highlighting crucial elements that require immediate attention or action.
>
> **Action Execution**: The LLM receives a textual description of the current game state and is prompted to decide on an action. It then outputs a text-based command, which our system interprets and executes in the game environment.
> The types of actions that can be executed include:
>
>   - Building construction (e.g., "Build Cybernetics Core")
>   - Unit production (e.g., "Train 5 Probes(protoss worker)")
>   - Research (e.g., "Research Blink Tech")
>   - Tactical maneuvers (e.g., "Scout with 1 probe", "Attack enemy base")
>
> For example, a command might be "Build 1 pylon(provide more supply)" or "Attack enemy base".
>
> For more detailed examples of these textual interactions, please refer to Appendix F: Policy Interpretability Examples.
>
> **Q3**: Deeper Insights
>
> In our experiments, we observed that optimizing LLMs through strategic prompting is essential for competitive performance against built-in AI opponents. Here are some crucial insights:
>
> Prompt Engineering: Effective prompts blend game mechanics with strategic concepts. For instance, prompts like "Consider resource management, tech tree progression, and army composition" led to better results than simpler directives. Summarized game state information at each decision point also enhanced performance.
>
> Prompt Sensitivity: LLMs showed high sensitivity to how decision scenarios are framed. Proactive prompts, such as "What potential threats should you prepare for?", generally led to better outcomes than reactive prompts.
>
> For example, in a test, GPT-3.5 with optimized prompts achieved an 84% win rate against the Level 4 built-in AI, significantly higher than the 12.5% win rate with basic prompts, largely due to improved resource management and anticipatory defensive strategies.

---

> > ### Comment · Reviewer_B3oZ · 2024-08-12
> >
> > Thanks for the rebuttals. My concerns are mainly addressed. I will raise my score to 6.

---

### Decision · Program_Chairs · 2024-09-25

**Decision:**

Accept (poster)

**Comment:**

The reviewers and I agree that this paper makes a solid contribution to benchmarking for LLMs. While the paper is light on theoretical contributions, it is an excellent piece of empirical work and will likely be of interest to a wide variety of researchers.

As a side note, I sincerely hope you're working on trying this with visual data using a multi-modal LLM!